# Disentangling spring-neap SPM dynamics in estuaries

Yoeri M. Dijkstra[1], Dennis D. Bouwman[2], and Henk M. Schuttelaars[1]

[1]Delft Institute of Applied Mathematics, Delft University of Technology, Delft, Netherlands
[2]Centrum voor Wiskunde en Informatica (CWI), Amsterdam, Netherlands

**Correspondence:** Yoeri M. Dijkstra (y.m.dijkstra@tudelft.nl)

**Abstract.** Suspended particulate matter (SPM) concentrations in estuaries have been observed to vary strongly over the spring-neap cycle through complex interactions between trapping and re-suspension. However, a systematic framework for analysing the processes causing this spring-neap SPM variability in general is missing. In this study we set-up such a framework, consisting of three tiers. First, by studying the sediment transport capacity, it is identified how the locations of sediment trapping change over the spring-neap cycle. Second, it is studied how the transport capacity affects the sediment stock and bottom pool of sediment. This bottom pool only adapts gradually to the changing transport conditions, incorporating a lag or memory effect. Using a two-timescales analysis it is shown this slow movement of the bottom pool is the leading source of such lag effects. Third, the SPM concentration is explained from an almost instantaneously balanced exchanged between the bottom pool and the water column through re-suspension and deposition.

We demonstrate the use of this framework on two model cases implemented in the idealised width-averaged iFlow model: an idealised test case where the sediment dynamics does not affect the water motion and a case representative of the Loire estuary, with strong feedback between sediment and the water motion through sediment-induced damping of turbulence. The first is illustrative as it allows a full understanding in terms of cause-and-effect between water motion, transport and SPM concentration. In the more realistic Loire case, the SPM dynamics cannot be explained in terms of cause and effect, but can explain the trapping locations and timing of maximum concentrations in a systematic way in terms of the governing physical mechanisms.

## 1 Introduction

In estuaries, suspended particulate matter (SPM) tends to concentrate in specific zones, called estuarine turbidity maxima (ETM). When assuming equilibrium conditions, ETM are often associated with sediment trapping, i.e. convergence of subtidal sediment transport capacity. This leads to the formation of a bottom pool of sediment. From this bottom pool, sediment is re-suspended to form the ETM (Burchard et al, 2018). However, since estuaries are highly dynamic environments, the sediment dynamics is often not in equilibrium, e.g. due to variations in flow on the spring-neap and seasonal timescales. Such flow variations affect the amount of re-suspension as well as the location and strength of sediment trapping. Moreover, if a sediment bottom pool was formed, it takes time to adapt to such changing flow conditions. Hence, the ETM may exist in regions without trapping of sediment, due to remnants of a bottom pool that was formed under past flow conditions (e.g. Brouwer et al, 2018; Schoellhamer, 2011).

Focussing on the spring-neap timescale, fortnightly variability of SPM concentrations has been observed in various estuaries including the Hudson (Traykovski et al, 2004), Seine (Le Hir et al, 2001), Tamar and Weser (Grabemann et al, 1997), Ems (Winterwerp et al, 2017), Scheldt (Fettweis et al, 1998) and Gironde (Allen et al, 1977). Observed dynamics has been attributed to various phenomena. Firstly, the spring-neap cycle affects the SPM concentration through re-suspension. Various authors report higher sediment concentrations during spring than neap, because higher bed shear stresses lead to more re-suspension (e.g. Allen et al, 1980; Vale and Sundby, 1987; Grabemann et al, 1997; Fettweis et al, 1998). Stronger vertical stratification during neap tides than spring in stratified estuaries reinforces this effect (Jay and Musiak, 1994). Secondly, the spring-neap variation affects the SPM concentrations through the trapping of sediments. In stratified estuaries where sediment trapping is dominated by density-driven flow, increased trapping of sediment is observed during neap tide. This is due to reduced tidal mixing, which causes stronger stratification and density-driven flow (e.g. Schoellhamer, 2000; Ralston and Geyer, 2009). Conversely, there are examples of estuaries with tide-dominated sediment trapping where trapping is strongest during spring tides (e.g. Uncles et al, 2006).

The summary above indicates there is a great amount of observational evidence of spring-neap SPM dynamics and various processes that affect both trapping and resuspension in complex ways. The way these various processes together result in the SSC is poorly understood. Underlying this knowledge gap, we observed that a framework to systematically assess spring-neap variations of the flow on trapping and resuspension and their combined effect on SSC is essentially missing.

The goal of this study is to gain insight into the complex interactions between sediment trapping, resuspension and SPM concentration under spring-neap variations of the flow. To this end, we developed a systematic framework to analyse and understand these interactions. Our framework will first be illustrated using an example of an idealised tide-dominated estuary, where trapping is dominated by tide-induced sediment transport processes. The effect of the spring-neap cycle on these trapping processes as well as re-suspension will be analysed in detail. By varying the erosion parameter for sediment in this test case, the effect of bottom pool formation on the final SPM dynamics is specifically emphasised. The model is then applied to a more realistic test case representing the hyperturbid Loire estuary, demonstrating the various effects of the spring-neap cycle on both tide- and baroclinically-induced transport processes in a more complex context. While it is not our aim to provide an extensive overview of the the various spring-neap-related processes that can occur in different types of estuaries, the framework developed can be used to systematically study other estuaries as well.

This method is implemented in an idealised width-averaged model based on the iFlow framework (Dijkstra et al, 2017; Brouwer et al, 2018). While our analysis is not strictly limited to idealised models, the iFlow model facilitates gaining more understanding of the various sediment transport processes that can later be projected to better interpret more complex model results or observations. The iFlow model allows studying a decomposition of sediment transport processes as well as facilitates a formal definition and analysis of bottom pool dynamics.

The outline of this paper is as follows: Section 2 introduces the model, analysis methods, solution method, and set-up of our case studies. The results for the idealised set-up and Loire case are then discussed in Section 3. A discussion of limitations and implications of this work is presented in Section 4, followed by the conclusions.

## 2 Model and methods

### 2.1 Model equations and forcing

We use a process-based width-averaged model that solves for the equations for water motion, sediment mass conservation and dynamics of the sediment bottom pool. The model domain is described by Cartesian $(x, z)$-coordinates. The estuary is assumed to be a single channel running from the mouth $(x = 0)$ to the upstream limit $(x = L)$, see Fig. 1. The vertical axis runs from an $x$-dependent bed level $z = -H(x)$ to the surface $z = R(x) + \zeta(x, t)$. Here, $R(x)$ is a subtidal reference level caused by the river set-up and $\zeta(x, t)$ is the (tidal) surface variation. The width of the estuary is also $x$-dependent and denoted by a function $B(x)$. The equations for the water motion are the width-averaged continuity and momentum equations. Assuming hydrostatic pressure and the Boussinesq approximation these equations read as

$$(Bu)_x + Bw_z = 0, \tag{1}$$

$$u_t + uu_x + wu_z = -g\zeta_x + g\beta \int_z^{R+\zeta} s_x \, dz' + (A_\nu u_z)_z. \tag{2}$$

Here, subscripts $x$, $z$, and $t$ denote derivatives with respect to these dimensions. The functions $u$ and $w$ are the velocity components in the $x$ and $z$ directions, $g$ is the acceleration of gravity, $\rho_0$ is a reference density, and $A_\nu$ is the vertical eddy viscosity. The function $s(x, z, t)$ denotes salinity and $\beta$ is the haline contraction coefficient; it is assumed that density differences are dominated by salinity. Specific choices for modelling salinity are discussed in Section 2.6. At the surface, a no-stress condition and kinematic condition are prescribed. At the bed, we prescribe the partial sip law $A_\nu u_z = s_f u$, where $s_f$ is a partial slip parameter, and a non-permeability condition. The partial slip parameter and eddy viscosity, both representing effects of turbulence, are assumed constant over the semi-diurnal cycle for simplicity (but potentially varying over the spring-neap cycle). The eddy viscosity is assumed vertically uniform.

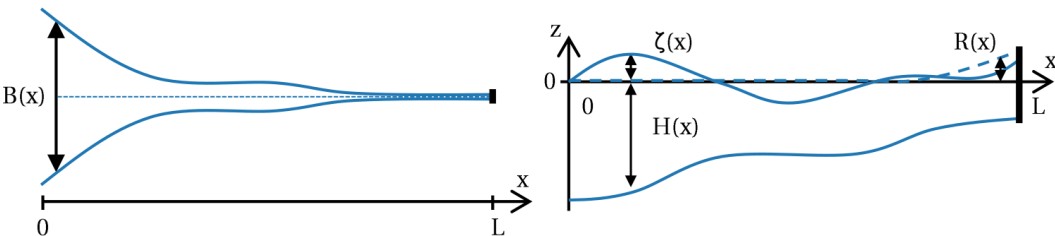

**Figure 1.** Model domain. The model is two-dimensional in along-channel $(x)$ and vertical $(z)$ direction and is width-averaged. The depth and width are allowed to vary smoothly with $x$. Figure copied from Dijkstra et al (2017).

The sediment mass balance is described by the equation

$$c_t + uc_z + (w - w_s)c_z = (K_\nu c_z)_z + (K_h c_x)_x, \tag{3}$$

where $c$ is the suspended sediment concentration, $w_s$ is the settling velocity, and $K_\nu$ and $K_h$ are the vertical and horizontal eddy diffusivities. Both $K_\nu$ and $K_h$ are vertically uniform and constant over the tidal cycle. At the surface, a no-flux condition is used. At the bottom, re-suspension of sediment is considered according to

85 $$K_\nu c_z = M|\tau_b|f \qquad \text{at } z = -H, \tag{4}$$

where $M$ is an erosion parameter, $\tau_b$ is the bed shear stress, and $f$ is the erodibility. The erodibility is a value between 0 and 1 indicating the amount of sediment available on the bed averaged over the tidal cycle, from permanently starved conditions at $f = 0$ to abundant supply at $f = 1$ (Brouwer et al, 2018).

The amount of sediment at the bed is modelled using the Exner equation, stating that the change of sediment mass on the
90 bed equals deposition minus re-suspension:

$$S_{\text{bed, t}} = w_s c|_{z=-H} - M|\tau_b|f, \tag{5}$$

where $S_{\text{bed}}$ is the amount of easily erodible sediment in the bottom pool, or *bottom sediment stock* (in kg/m$^2$). It is assumed that the thickness of the bottom pool does not affect the water depth. It is useful to also define the total tidally-averaged sediment stock $S$, which describes the amount of easily erodible sediment in the bed and water column, i.e.

$$S = \left\langle S_{\text{bed}} + \int_{-H}^{R+\zeta} c \, dz \right\rangle, \tag{6}$$

where $\langle \cdot \rangle$ denotes averaging over a typical semi-diurnal period (formally defined in Sect. 2.2.2). We relate the erodibility $f$ to the stock $S$ using the relation derived by Brouwer et al (2018). For small values of $S$, this relation states that $f$ increases with $S$ (i.e. supply limited conditions). For sufficiently large values of $S$, the erodibility equals unity, meaning that adding more sediment to the bottom pool will not affect the amount of sediment in suspension (i.e. erosion limited conditions).

The hydrodynamic forcing for this model consists of an $M_2$, $M_4$, $S_2$ and $S_4$ tide at the mouth (see also Sect. 2.2.2) and a constant discharge at the head of the estuary. For the sediment model a constant tidally-averaged and depth-averaged sediment concentration is prescribed at the mouth and we impose no inflow of sediment from the watershed. We do not consider any initial condition here, since we will only study the model in dynamic equilibrium, i.e. the state reached after a long time with concentrations varying over the semi-diurnal and spring-neap timescale but not changing on sub-spring-neap timescales.

## 2.2  Solution methods

The solution method consists of several steps. First using a scaling and perturbation approach, the model equations are approximated following work by Dijkstra et al (2017) and Chernetsky et al (2010). Next, a scaling of the tidal and spring-neap dynamics will justify use of a two-timescales perturbation method to further approximate and simplify the temporal dynamics. The use of this method to analyse spring-neap dynamics is new, and it will be demonstrated that it not only leads to numerically
efficient solutions, but also facilitates understanding of the dynamics. The resulting equations are solved numerically

### 2.2.1 Perturbation approach

The model equations (1)-(5) are solved within the iFlow modelling framework using a perturbation approach. This means that the equations are solved by approximation under the assumption that the ratio of the tidal amplitude and the depth at the mouth equals $\epsilon$, with $\epsilon \ll 1$. Also, bathymetric and geometric variations are assumed to be present only on the length scale of the estuary. The perturbation approach allows for semi-analytical solutions as well as decomposition of the sediment transport according to the physical process causing it, see Chernetsky et al (2010) or Dijkstra et al (2017) for details.

### 2.2.2 Tides - a two-timescales perturbation approach

The forcing by the lunar ($M_2$ and $M_4$) and solar ($S_2$ and $S_4$) tides results in a relatively fast semi-diurnal and quarter-diurnal tidal motion, modulated by a slowly varying amplitude and phase over the spring-neap cycle, as illustrated in Fig. 2. To disentangle the dynamics happening at these strongly different timescales we use a two-timescales perturbation method (e.g. Holmes, 2013). Below we illustrate the use of this method and focus on how this method helps to understand the physics.

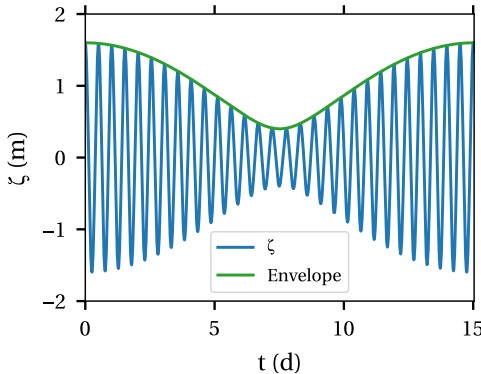

**Figure 2.** Example of the tidal elevation caused by superposition of the $M_2$ and $S_2$ tide during a spring-neap cycle (blue) and the resulting envelope function (green).

To illustrate the method, we focus on the $M_2$ and $S_2$ water level. We write these components of the water level $\zeta$ as

$$\zeta(x,t) = \Re \left( \hat{\zeta}_{M_2}(x) e^{i\omega_{M_2} t} + \hat{\zeta}_{S_2}(x) e^{i\omega_{S_2} t} \right), \tag{7}$$

where $\hat{\zeta}_{M_2}$, $\hat{\zeta}_{S_2}$ are the complex phase-amplitudes corresponding to the $M_2$ and $S_2$ tide, and $\omega_{M_2} = 1.405 \cdot 10^{-4}$ s$^{-1}$ and $\omega_{S_2} = 1.454 \cdot 10^{-4}$ s$^{-1}$ are the corresponding angular frequencies. We define the typical timescale $\tau = \omega_{M_2} t$ and re-order and

rewrite E. (7) to

$$\zeta(x,t) = \Re\left(\underbrace{\left[\hat{\zeta}_{M_2}(x) + \hat{\zeta}_{S_2}(x)e^{i\frac{\omega_{S_2}-\omega_{M_2}}{\omega_{M_2}}\tau}\right]}_{\hat{E}(x,t)}e^{i\tau}\right),$$ (8)

where the expression $\hat{E}(x,t)$ is the slowly varying envelope function (green line in Fig. 2). We observe two dimensionless timescales: $\tau_1 = \tau = \omega_{M_2}t$ and $\tau_2 = \frac{\omega_{S_2}-\omega_{M_2}}{\omega_{M_2}}\tau$. The factor $\delta \equiv \frac{\omega_{S_2}-\omega_{M_2}}{\omega_{M_2}} \approx 0.035$ is much smaller than unity, and hence $\tau_2$ will be referred to as a slow timescale, related the envelope in Fig. 2, and $\tau_1$ is referred to as a fast timescale. Correspondingly, we define two dimensional time variables $t_1 = \tau_1/\omega_{M_2} = t$ and $t_2 = \delta t$. Rewriting Eq. (8) in terms of these two time variables we see that

$$\zeta(x,\tau_1,t_2) = \Re\left(\underbrace{\left[\hat{\zeta}_{M_2}(x) + \hat{\zeta}_{S_2}(x)e^{i\omega_{M_2}t_2}\right]}_{\hat{E}(x,t_2)}e^{i\omega_{M_2}t_1}\right),$$ (9)

and the envelope function $\hat{E}(x,t_2)$ only depends on the slow time variable $t_2$. All time derivatives occurring in the governing equations are rewritten in terms of $t_1$ and $t_2$ as

$$\frac{\partial}{\partial t} = \frac{\partial}{\partial t_1} + \delta\frac{\partial}{\partial t_2}.$$ (10)

We now introduce the main assumptions of the two-timescales perturbation approach:

1. $t_1$ and $t_2$ are considered *independent* variables. This means that $t_1$ can be varied while keeping $t_2$ constant and vice-versa.

2. Terms in the equations of order $\delta$ can be neglected when compared to terms of order 1. Hence, to leading order, a time derivative $\frac{\partial}{\partial t}$ is approximated as $\frac{\partial}{\partial t_1}$.

3. The small variable $\delta$ is considered $\mathcal{O}(\epsilon^2)$, where $\epsilon \ll 1$ is the parameter in the perturbation method discussed in the previous section.

By the decoupling of the time variables $t_1$ and $t_2$, $t_1$ no longer strictly represents the $M_2$ tide but rather a typical semi-diurnal tide that combines the $M_2$ and $S_2$ tide. We will denote this as a $D_2$ tide. Likewise the quarter-diurnal tide represents the combined effects of the $M_4$. $S_4$, and $MS4$ tides and is referred to as the $D_4$ tide. Here, we approximate the period of the $D_n$ tidal component ($n = 1, 2, \ldots$) by the period of the corresponding lunar component $M_n$. Tidal averaging, denoted by $\langle \cdot \rangle$ hence is defined formally as averaging over the time variable $t_1$, i.e. averaging over the $D_2$ tidal component.

Applying these assumptions to the governing equations and combining them with the scaling as detailed in Chernetsky et al (2010) and Dijkstra et al (2017), we find the following. To leading order and order $\epsilon$, all terms involving $\delta\frac{\partial}{\partial t_2}$ are neglected. For both the water motion and sediment dynamics, the model result only accounts for terms of leading order and order $\epsilon$, since these form the dominant balances. Hence, the water motion is in dynamic equilibrium at each stage of the spring-neap cycle.

In other words, a changing value of the envelope function $\hat{E}(0, t_2)$ at the mouth is transferred immediately to the water motion in the entire estuary. Similarly, the sediment distribution over the water column in dynamic equilibrium at each stage of the spring-neap cycle; i.e. when varying $t_2$, the vertical sediment distribution is assumed to adapt immediately to the changing hydrodynamics.

However, the distribution of sediment in the bottom pool $S_{\mathrm{bed}}$ and stock $S$ do not adapt instantly when varying $t_2$, but adapt slowly to the changing conditions. To see this, we first derive an equation for the time-evolution of $S$. This is done by integrating Eq. (3) over the water column and combining this with the Exner equation, Eq. (5). Tidal averaging then yields (Brouwer et al, 2018)

$$BS_t = -\left\langle B \int_{-H}^{\zeta} uc - K_h c_x \, dz \right\rangle_x . \tag{11}$$

Applying our two-timescales assumptions, we see that the tidally-averaged stock $S$ depends only on $t_2$, not $t_1$ and the equation rewrites to

$$B\delta S_{t_2} = -\left\langle B \int_{-H}^{\zeta} uc - K_h c_x \, dz \right\rangle_x . \tag{12}$$

Using a scaling analysis (see Chernetsky et al (2010)), it can be shown that the right-hand side is at maximum of order $\epsilon^2$. Since $\delta$ is of order $\epsilon^2$, both sides of the equations balance. Physically, this means that the sediment stock varies gradually on the slow timescale.

This equation clearly demonstrates the conceptual understanding obtained by using the two-timescale approach. Eq. (12) shows that the change in the stock over the slow timescale is a function of the divergence of the total tidally averaged sediment transport. Hence, the effects of spring-neap variations can be fully understood by analysis of the tidally-averaged transports and the evolution of the stock on the slow timescale. Additionally, without the two-timescales approach, 'tidally-averaged' is ambiguous and one would need to choose to average over e.g. the lunar semi-diurnal cycle. Finally, as will be argued below, the method allows for a more efficient numerical solution method and hence much reduced computation times.

## 2.3 Analytical and numerical solution procedures

The vertical velocity and sediment profiles are resolved using semi-analytical methods (i.e. analytical supplemented by numerical quadrature rules where needed, see Dijkstra et al (2017)). The along-channel dimensions are discretised using a finite difference method of first-order for advective terms and second-order for dispersive terms on an equidistant grid. In the fast time variable $t_1$, the model is solved in terms of harmonic components. As the forcing consists only of residual, $D_2$ and $D_4$ components, the solution consists of residual, $D_2$, $D_4$ and overtidal components. Using harmonic components means that the model is directly solved for dynamic equilibrium in the variable $t_1$; i.e. no time-stepping or spin-up is needed. In the slow $t_2$ time variable, the model is solved using time-stepping, starting from an initial condition for $S$. In this study, we start from $S = 0$ and time-integrate until a dynamic equilibrium on the spring-neap cycle is reached. We used 200 time steps per spring-neap cycle, implying a time step of approximately 1.8 hours. Note that this timestep is much coarser than what would usually

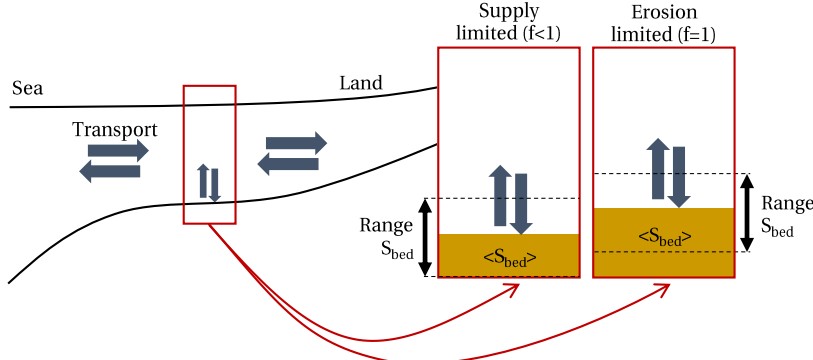

**Figure 3.** Conceptual depiction of SPM dynamics in an estuary, showing subtidal along-channel transport and the exchange in a water column. Two cases are shown for the water column: supply limited and erosion limited. In the supply limited state, the bottom pool is empty during part of the semi-diurnal cycle, while there is always erodible sediment present in the erosion limited state.

be needed if time-stepping were also used for solving for the semi-diurnal tides. Hence, the application of the two-timescales approach results in much smaller computation times than would be needed otherwise.

## 2.4 Analysis method

In order to analyse the sediment dynamics, we study three processes: the $D_2$-tide averaged horizontal transport, bottom pool movement and re-suspension in the water column, as sketched conceptually in Fig. 3. These processes, of course, are coupled, yet it is useful to study them separately to better understand the SPM concentration.

First we consider the $D_2$-tide averaged horizontal transport. We analyse this using the *sediment transport capacity*: the transport that would occur if a small uniform layer of sediment were added on the bed. We look specifically for the locations where transport capacity converges, i.e. locations where the transport capacity changes from positive (i.e. importing) to negative (i.e. exporting). This means sediment tends to accumulate there. To further understand why the transport capacity changes over the spring-neap cycle, we will make a decomposition into contributions from various physical components. Here we will use the decomposition as implemented in iFlow, of which a selection of relevant terms is discussed below:

- *River*: Transport related to river flushing of tidally resuspended sediment and effects of tide-river interaction on sediment resuspension.

- *External $M_4$ tide*: Transport related to the tidal asymmetry caused by the interaction between the $D_2$ and $D_4$ tide propagating from the mouth in either resuspension or flow.

- *Internal $M_4$ tide*: Transport related to the tidal asymmetry caused by the interaction between the $D_2$ and the subtidal flow and $D_4$ tide generated inside of the estuary. Various processes generate subtidal flow and $D_4$ tide inside of the estuary, including tidal return flow, nonlinear momentum advection and velocity-depth asymmetry (representing that the vertical velocity profiles are not exactly equal during ebb and flood, because the depth of the estuary is different).

- *Spatial settling lag*: Transport due to spatial settling lag: the tendency for sediment to move to areas with minimum velocity amplitude.

- *Baroclinic contribution*: Transport related to the flow and resuspension caused by gravitational circulation due to the along-channel salinity gradient.

Second, the global $D_2$-tide averaged transport leads to changes in the sediment stock $S$ (i.e. sediment in suspension and at the bottom) at each location in the estuary. Associated with each stock is an erodibility $f$ used as measure for the bottom pool thickness. Changes in the stock occur on the slow time scale and hence it is needed to consider the history of the system. We
compare the bottom pool with the convergence points. If the system responds fast, the bottom pool will be most thick at the convergence points, while a delay may be present in slower systems.

Third, knowing the spatial distribution of sediment on the bottom, we may focus on each location of the estuary in isolation. The SPM concentrations at each location follows from the *local* balance between re-suspension and settling dynamics in the water column and interaction with the bottom pool. This can be expressed as (adapted from Chernetsky et al, 2010; Brouwer
et al, 2018)

$$c(t_1, t_2) = \hat{c}(t_1, t_2) f(t_2), \tag{13}$$

where $\hat{c}$ is the sediment capacity. The sediment capacity is the maximum concentration one could attain at any moment given there is sufficient sediment available (i.e. $f = 1$). It depends on hydrodynamic quantities (i.e. bed shear stress, turbulence) and sediment parameters (settling velocity, erosion parameter). We hence see that the SPM concentration follows from under-
220 standing both the sediment capacity and erodibility. Under supply limited conditions, $f < 1$, the bottom pool is so thin that all sediment is suspended from it for at least part of the semi-diurnal cycle (illustrated in the left water column in Fig. 3). Hence, both changes in the stock and changes in sediment capacity affect the SPM concentration. Under erosion limited conditions, $f = 1$, there is always sediment present on the bed (illustrated in the right water column in Fig. 3). Sediment added to the stock on a tide-averaged basis will go to the bottom pool without affecting $f$. During times of erosion limited conditions, the
225 observed SPM concentration is therefore not affected by horizontal transport but only by changing sediment capacity.

Overall, our framework consists of analysing how transports distribute the sediment stock along the estuary, after which we can look at the slowly varying bottom pool (via $f$) at each location. The SPM concentration then follows from an instantaneous balance between the bottom pool and the water column depending on the sediment capacity. We argue that this is a useful way of thinking about the SPM dynamics, but also acknowledge that transport, bottom pool and SPM concentration are mutually
dependent and this analysis does not imply the dynamics is linear.

## 2.5 Quasi-stationary and dynamic model experiments

We specifically want to highlight the importance of the slow timescale adaptation of the sediment stock. To this end, we compare the model results to their quasi-stationary equivalent. The model as described above accounts for the time it takes for the sediment stock in the estuary to adapt to the changing hydrodynamic conditions and is referred to as the *dynamic*

computation. This is compared to a case where the sediment stock is assumed to instantaneously adapt to the hydrodynamic conditions, referred to as *quasi-stationary*. The quasi-stationary case is a useful reference, because the sediment distribution is a direct consequence of the present hydrodynamic conditions without any memory effects. The mathematical procedure for computing quasi-stationary conditions is discussed in Appendix A. The analysis framework presented in the previous section will be applied to both the dynamic and quasi-stationary model runs to help highlight the importance of memory effects.

## 2.6 Design of model experiments

### 2.6.1 Idealised example

We use an idealised example to provide insight into the various interacting effects of the spring-neap cycle on the sediment dynamics. This example does not directly represent a particular estuary, but is configured to some of the properties of the Ems estuary to make sure we are in a realistic and practically relevant parameter space. The model domain is 64 km long, has a depth $H$ varying smoothly between $H = 10.5$ m at the mouth and 5 m at the head and a width $B$ according to $B = 800e^{-x/30000}$, where $x$ is the along-channel distance in metres. The tidal forcing parameters are inspired on a T-tide analysis (Pawlowicz et al, 2002) of the observed tidal data at the tidal station of Knock in the Ems. Salinity is prescribed according to

$$s = \frac{s_{\text{sea}}}{2}\left(1 - \tanh\left(\frac{x - x_c}{x_L}\right)\right), \tag{14}$$

where $s_{\text{sea}}$, $x_c$ and $x_L$ are fixed parameters and salinity is assumed constant over the tide and spring-neap timescale. The parameters are fixed to keep the analysis as simple as possible for this case. Similarly, the roughness parameter, eddy viscosity and eddy diffusivity are constant in time and space and independent of the velocity or sediment concentrations for simplicity. Values of the salinity and turbulence parameters were taken from Chernetsky et al (2010) and are roughly representative of the Ems estuary. It should be noted though that the Ems features very high sediment concentrations where sediment-turbulence feedbacks are important. Such effects are ignored here for simplicity. As a consequence, the obtained results are not representative of the observed dynamics in the Ems. All parameter values are shown in Table 1. The model is solved analytically in the vertical direction and numerically in the horizontal direction. The horizontal grid contains 200 cells, and the time integration uses seven spring-neap cycles for spin-up and an eighth cycle for analysis. This is sufficient to reach dynamic equilibrium.

### 2.6.2 Loire case study

To further illustrate the effects of spring-neap variations in a more realistic setting, we will discuss an application to the Loire estuary. The Loire is a hyperturbid estuary in France, featuring sediment concentrations well over 10 g/l during low discharge conditions. The spring-neap differences are very pronounced, with the modelled $D_2$ tidal amplitude at the mouth varying between 1.1 m and 2.4 m, and large differences in observed sediment concentrations over the spring-neap cycle (e.g. Jalón-Rojas et al, 2016). We will use the schematisation and iFlow model of Dijkstra and De Goede (Submitted to Journal of Geophysical Research: Oceans) (*hereafter DdG24*).

| | Parameter | Value | | Parameter | Value |
|---|---|---|---|---|---|
| $A_{M_2}$ | $M_2$ amplitude at $x=0$ | 1.39 m | $A_{M_4}$ | $M_4$ amplitude at $x=0$ | 0.17 m |
| $A_{S_2}$ | $S_2$ amplitude at $x=0$ | 0.35 m | $A_{S_4}$ | $S_4$ amplitude at $x=0$ | 0.013 m |
| $\phi_{M_2}$ | $M_2$ phase at $x=0$ | 335 deg | $\phi_{M_4}$ | $M_4$ phase at $x=0$ | 138 deg |
| $\phi_{S_2}$ | $S_2$ phase at $x=0$ | 47 deg | $\phi_{S_4}$ | $S_4$ phase at $x=0$ | 356 deg |
| $Q$ | River discharge | 45 m$^2$/s | $w_s$ | Sediment settling velocity | 2 mm/s |
| $A_\nu$ | Eddy viscosity | 0.012 m$^2$/s | $c_{\text{sea}}$ | Depth-averaged tide-averaged sediment concentration at $x=0$ | 20 mg/l |
| $K_\nu$ | Eddy diffusivity | 0.012 m$^2$/s | $s_{\text{sea}}$ | Seaward salinity | 30 psu |
| $s_f$ | Partial slip parameter | 0.049 m/s | $x_c$ | Salinity parameter, Eq. (14) | -3.5 km |
| $K_h$ | Horizontal eddy diffusivity | 100 m$^2$/s | $x_L$ | Salinity parameter, Eq. (14) | 11 km |
| $M$ | Erosion parameter | $1.5 \cdot 10^{-4}$ s/m | | | |

**Table 1.** Default values of the parameters in the idealised case.

In this set-up, salinity is modelled dynamically following the model of MacCready (2004), resolving transport by gravitational circulation and tidal dispersion as well as computing vertical salt stratification. The salt is assumed to be in quasi-stationary state, i.e. adapting immediately to changing spring-neap conditions. The roughness parameter, eddy viscosity and eddy diffusivity are still vertically uniform and time-independent but their values depend dynamically on the velocity, depth and salt- and sediment stratification. Furthermore, the settling velocity $w_s$ is not constant but accounts for the effects of hindered settling. Hence, the effects of the spring-neap cycle on salinity and turbulence, as well as density-driven flow are taken into account. The model is applied for a constant representative summer discharge of 250 m$^3$/s and tidal characteristics are derived from a T-tide analysis of observations at the estuary mouth. For all model settings, model calibration and validation, we refer to DdG24. Model settings and a summary of the turbulence closure model are also included in Appendix B. Whereas DdG24 assumed different values for $c_{\text{sea}}$ for their spring and neap cases, we will use their neap tide value of $c_{\text{sea}} = 0.3g/l$ for the entire simulation.

## 3 Results

### 3.1 An idealised case study

Fig. 4 shows results of the idealised case study, comparing the quasi-stationary results with the dynamic results. Fig. 4a shows the spring-neap averaged along-channel near-bed concentration for the quasi-stationary case (blue line) and dynamic case (red line). The shaded areas indicate the range of variation of the semi-diurnally-averaged (i.e. $D_2$-averaged) near-bed concentration over the spring-neap cycle. Looking at the mean concentrations, ETM are observed near km 8 and 55 in both the quasi-stationary and dynamic case. Moreover, in this specific case it turns out that mean concentrations are of a similar magnitude. This is not generally the case as will be illustrated in the following sections. A generally valid result is that the

spring-neap variation of the $D_2$-averaged concentration is smaller in the dynamic case than in the quasi-stationary case. Fig. 4b-c show the near-bed concentration as a function of $x$ and $t$. In the quasi-stationary case, the highest concentrations are attained after spring tide near km 55 (marked by x in the figure) and the ETM moves gradually downstream towards neap. The ETM is absent in the time between neap and spring tide. In the dynamic case, the ETM near km 55 remains present during the entire spring-neap cycle and moves by about 10 km during this time. Interestingly, another peak in concentration is observed some time after neap tide near km 45.

## 3.2 Analysis

To better understand the dynamics we apply the analysis framework developed in Section 2.4. Hence, we first look at the transport capacity. Since there is no feedback of the sediment stock or concentration to the water motion and sediment capacity in this simple model, the transport capacity depends on the instantaneous flow conditions and hence is the same in the quasi-stationary and the dynamic case. The transport capacity scales with the bed friction, which, in this model, scales linearly with the $D_2$ velocity amplitude near the bed. To facilitate comparison of the transport capacity during spring and neap tide, we therefore divide the transport capacity by the near-bed velocity amplitude at $x = 0$. Fig. 5 shows this rescaled transport capacity for the default case at neap tide (Fig. 5a) and spring tide (Fig. 5b). First looking at the total transport capacity, we see two sediment convergence zones (i.e. transitions from importing capacity to exporting capacity): at km 15 and 41 during neap tide and km 5 and 56 during spring tide. During neap and spring tide, the transport capacities due to the river flow are equal after scaling by the near-bed velocity amplitude. The transport capacity due to the external $D_4$ tide is almost the same after scaling, while the contributions related to the internally generated $D_4$ tide and spatial settling lag are much larger during spring than neap. These dependencies of the transport components on the stage of the spring-neap cycle can be explained from analysis of the model equations. Results are presented in Table 2. The transport capacity depends on the tidal velocity, which (due to linearity) scales with the tidal surface elevation amplitude $|A_{D_2}|$ and $|A_{D_4}|$ (i.e. $D_2$ and $D_4$ tidal elevation) and a function of the local $D_2$-$D_4$ phase difference $f\left(\phi_{D_4} - \phi_{D_2}\right)$ (see Bouwman (2019) for an explicit expression of this function). Since the transport capacity due to river discharge scales linearly with the $D_2$ amplitude (and hence the $D_2$ velocity), the *rescaled* transport capacity due to river discharge in Fig. 5 is constant over the spring-neap cycle. The transport capacity by the externally generated $D_4$ tide is related to the asymmetry between the linearly propagating parts of the $D_2$ and $D_4$ tides, and also scales with the $D_2$ amplitude and additionally the variation of the $D_4$ amplitude and $D_2 - D_4$ phase difference are small between spring and neap in this particular case. Hence also the *rescaled* transport capacity of this component is similar during both phases. The internally generated $D_4$ tide is caused by the quadratic interaction of the $D_2$ tide due to processes including advection and stokes drift. Hence, the transport capacity due to the internally generated $D_4$ tide relates to the asymmetry between this nonlinearly generated $D_4$ tide and the $D_2$ tide and scales with the tidal amplitude cubed. Similarly, spatial settling lag follows from nonlinear interactions and scales with the tidal amplitude cubed. This explains the difference between neap and spring transport capacities.

Secondly, we study the variation of the bottom pool. Fig. 4d-e show the erodibility in the quasi-stationary and dynamic cases. The black lines indicate the location of the sediment convergence points in the quasi-stationary case (leading to either a local

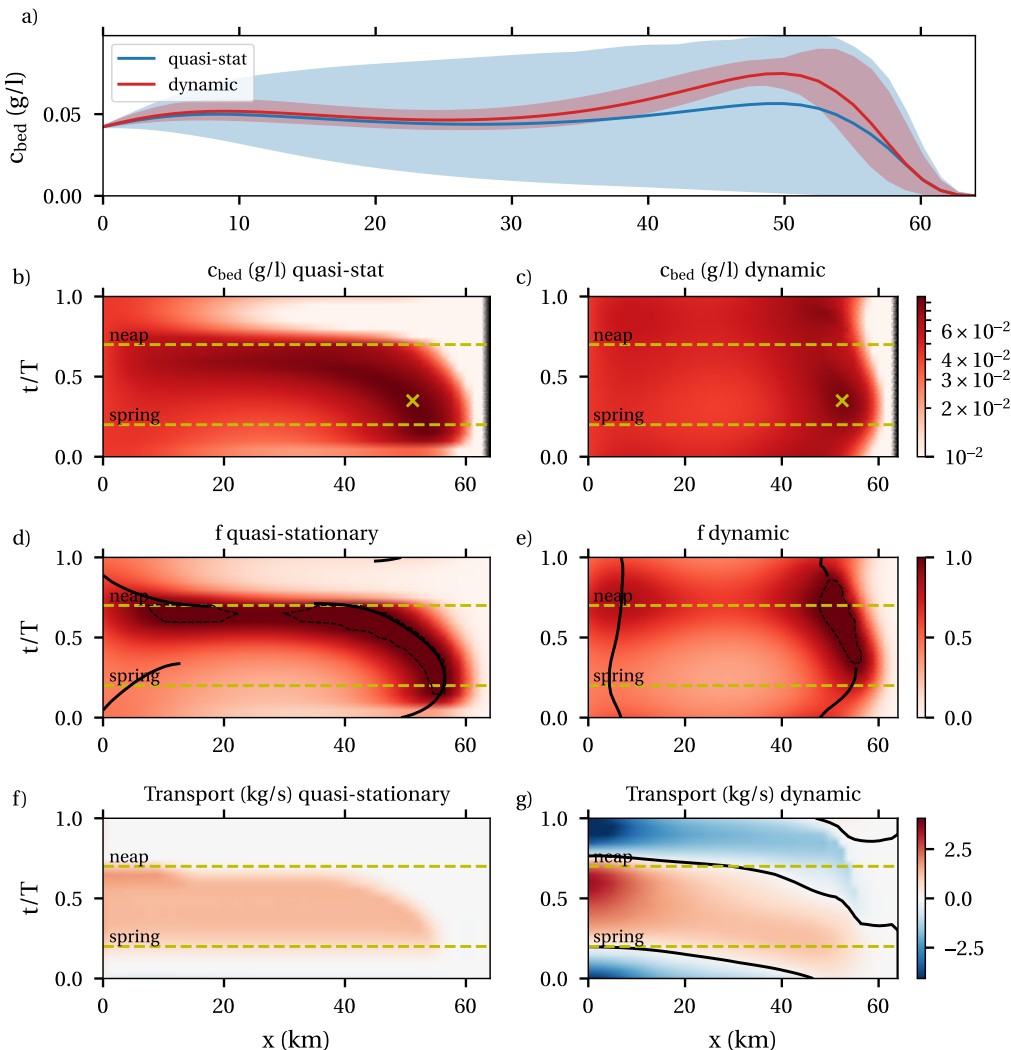

**Figure 4.** Results for the sediment concentration in the default case, comparing quasi-stationary and dynamic results. a) mean near-bed concentration (lines) and the spring-neap range of the tidally-averaged near bed concentration (shaded area). b-c) near-bed concentrations in the quasi-static and dynamic cases as function of $x$ and $t$. The x marks the maximum concentration. d-e) erodibility as a function of $x$ and $t$. In d) black solid lines indicate the convergence of transport capacity. In e) the black solid lines indicate local maxima. The black dashed lines indicate areas where $f = 1$. f-g) sediment transport as function of $x$ and $t$

maximum in erodibility or $f = 1$) and the local maxima in erodibility in the dynamic case. The dashed lines indicate areas where erosion limited conditions ($f = 1$) are found. In the quasi-stationary case, convergence points are found during part of the cycle between km 0 and 10 as well as further upstream. At spring and neap tide (yellow dashed lines), the convergence points match the downward zero-crossings of the transport capacity in Fig. 5. Erosion limited conditions are found going from

spring to neap tide, directly downstream from a sediment convergence point. In the dynamic case, local maxima in erodibility are found near km 5-8 and 45-55. An erosion limited zone is found from late spring to neap near km 50. Comparing the dynamic and quasi-steady case, it is observed that erosion limited conditions are not immediately attained at spring tide in the dynamic case, since it takes time to build up the bottom pool. This is confirmed by the transport (Fig. 4g), which shows that sediment is imported towards the bottom pool around spring. Similarly, erosion limited conditions do not immediately disappear after neap tide. The transport shows sediment is exported, but it takes time for sediment to leave the bottom pool. Also, the bottom pool moves much less in the dynamic case compared to the quasi-stationary case. Just before neap in the quasi-stationary case, the ETM moves downstream in a short time, while this downstream movement is only weakly present in the dynamic case. After neap in the quasi-stationary case, the ETM disappears. In the dynamic case we do observe exporting of sediment, but this is not fast enough for the ETM to disappear entirely. Note that transports in the quasi-stationary case are either positive, indicating there are erosion limited conditions with a growing bottom pool, or transport is zero.

Finally, we will look at the SPM concentration, which is the product of the sediment capacity and erodibility. In this idealised model, sediment capacity $\hat{c}$ changes only through variations in the bed shear stress over the spring-neap cycle. Hence, the subtidal sediment capacity is highest during spring tide and smallest during neap tide. In the quasi-stationary case (Fig. 4b), the highest concentrations are found just after spring tide. There are erosion limited conditions and hence concentration depends on the bed shear stress. Whereas the bed shear stress is maximum at spring tide, the bottom pool is slightly further downstream after spring tide, where bed shear stresses are higher than upstream. During the spring-neap cycle, the ETM follows the location of the maximum erodibility. In the dynamic case (Fig. 4c), the location of the highest concentrations follows the location of the bottom pool. Now, highest concentrations occur some time after spring tide not only because of the bed shear stress, but because the bottom pool still needs time to build up. Another peak in concentration is found after neap tide, as the bottom pool is still present for some time while the estuary is still exporting sediment and bottom shear stresses increase again.

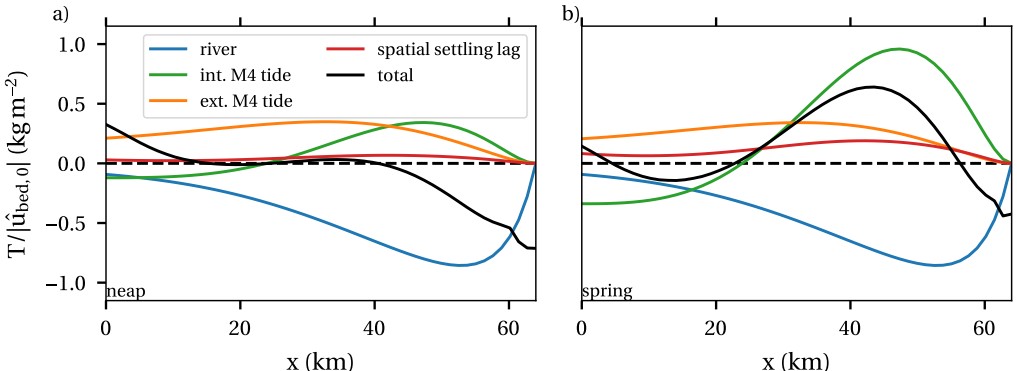

**Figure 5.** Transport capacity (black) and the leading contributions to the transport capacity for neap and spring. These are the same for the quasi-stationary and dynamic cases.

| Transport contribution | Scaling with $D_2$ and $D_4$ tide |
|---|---|
| River | $|A_{D_2}|$ |
| Ext. $D_4$ tide | $|A_{D_2}||A_{D_4}|f(\phi_{D_4} - \phi_{D_2})$ |
| Int. $D_4$ tide | $|A_{D_2}|^3$ |
| Spatial settling lag | $|A_{D_2}|^3$ |

**Table 2.** Scaling of some transport components with the $D_2$ and $D_4$ tidal amplitude and phase ($\phi$). Note that these dependencies are valid in the context of the model for the idealised case. Various factors are not taken into account in these dependencies, including the scaling of turbulence parameters with the tidal amplitude and effects of salt and SPM concentration on turbulence.

### 3.3 Effect of bottom pool presence

The timing and location of the ETM resulting from spring-neap variations depend strongly on whether or not a bottom pool with erosion limited conditions ($f = 1$) can form. If such a bottom pool forms, sediment keeps accumulating and the sediment concentration is at capacity conditions. To show the effect of the bottom pool on the SPM concentration we compare cases with different values of the erosion parameter $M = 1 \cdot 10^{-5}$ s/m and $M = 1 \cdot 10^{-3}$ s/m. Within this model, the sediment transport capacity simply scales linearly with the erosion parameter, while the spatial distribution remains the same. Hence,

the underlying sediment transport tendency remains the same as in the default case (cf. Fig. 5).

Fig. 6a, c, e, g show the sediment concentration, erodibility and transport for $M$ small. All results correspond to the dynamic case, with the only comparison with the quasi-stationary case in panel a. Due to the small erodibility, maximum concentrations are quite small, but since the transport capacity is the same, ETM are still found in the same locations (near km 5 and 50). The mean concentration in the dynamic case is bigger than that in the quasi-stationary case (panel a), while the variation is smaller.

The mean concentration is higher, because the sediment concentration is at capacity conditions ($f = 1$, panel e) in areas at the entrance and near km 50 during the entire spring-neap cycle. Sediment is still imported and exported to and from the bottom pool (panel g), but erosion limited conditions persist. Hence, the ETM does not vanish after neap as in the quasi-steady case (cf. Fig. 4c). Consistent with the capacity conditions, the highest concentrations in both ETM are found during spring tide, when bed shear stresses are at their maximum.

Fig. 6b, d, f, h show the results for the larger value of $M$, again corresponding to the dynamic case. The ETM near km 5 is still present but only very weak, while there is still a clear ETM near km 50. Mean concentrations in the dynamic case are significantly smaller than in the quasi-stationary case, while variations over the cycle remain smaller in the dynamic case (panel b). The estuary never reaches capacity conditions ($f < 1$) and maximum erodibility in both ETM is reached close to neap tide (Fig. 6f). Maximum concentrations are now found between spring and neap tide, when erodibility is approaching its maximum,

while the bed shear stress is not yet at its minimum. After neap tide, the ETM does not disappear as in the quasi-steady case, but concentrations do decrease after neap tide. This is because the bottom pool is not big enough to remain present for long after neap tide, preventing increasing concentrations even though the bed shear stress increases.

Comparing the situations with high and low $M$, it is observed that there is a significant dynamic effect of the spring neap cycle in both cases, compared to the quasi-stationary state. In both cases, it takes significant time for the estuary to move the sediment stock around and adapt to the changing conditions. Under erosion limited conditions, an important dynamic effect is the build-up and break-down of the bottom pool. During times of export, this means that erosion limited conditions can be sustained for a significant amount of time or even the entire spring-neap cycle as it takes time before the bottom pool is drained. This is reflected in a different timing of maxima and minima in the sediment concentrations. Indeed, comparing Fig. 6c and 6d, maximum concentrations for small $M$ occur at the time when minimum concentrations are attained for large $M$ and vice versa. Since the underlying transport capacity is identical in both cases, this is due to the bottom pool thickness relative to the possible re-suspension by the tide.

### 3.4 Adjustment time

In previous studies of the effects of spring-neap tides on salt distributions in estuaries, the dynamic effects of the spring-neap cycle are often quantified by measuring the delay between the salinity distribution and the tidal amplitude (e.g. Hetland and Geyer, 2004). For sediment, even in our highly simplified model, such relation is not possible. Fig. 5 already shows the complicated relation between the tidal amplitude and transport capacity. As the various physical contributions scale differently with the tidal amplitudes and $D_2$-$D_4$ phase difference, the spring-neap variation in total sediment transport capacity is not a monotone function of the tidal amplitude. This total transport capacity determines the tendency for the estuary to redistribute sediment over time which then results in an erodibility $f$ and finally the sediment concentration. As there is no monotone relation between the spring-neap forcing and the transport capacity, there is no monotone response of the sediment concentration to the tidal amplitude and hence analysis in terms of delay is not possible.

We may however still estimate a timescale for adjustment of the estuary to changing forcing conditions. If the adjustment time is much shorter than the spring-neap period, the dynamic SPM concentration will be similar to the quasi-steady result. If the adjustment time is similar to or bigger than the spring-neap period, results may deviate significantly from the quasi-stationary results as shown in the previous sections. A simple expression to estimate the order-of-magnitude of the adjustment time of the estuary $T_{\mathrm{adj}}$ could be:

$$T_{\mathrm{adj}} = \frac{\max_t \left( \int_0^L BS\,dx \right) - \min_t \left( \int_0^L BS\,dx \right)}{< |T| >_{x=0}}, \tag{15}$$

i.e. the difference between the maximum and minimum total amount of sediment in the estuary divided by the mean ($< \cdot >$) absolute sediment transport at the mouth. Note that this timescale is diagnostic, i.e. it requires knowledge of the sediment balance to compute it. By this definition, $T_{\mathrm{adj}}$ in the default case is approximately 23 days. This is longer than the spring-neap cycle duration, consistent with the results showing significant dynamic effects.

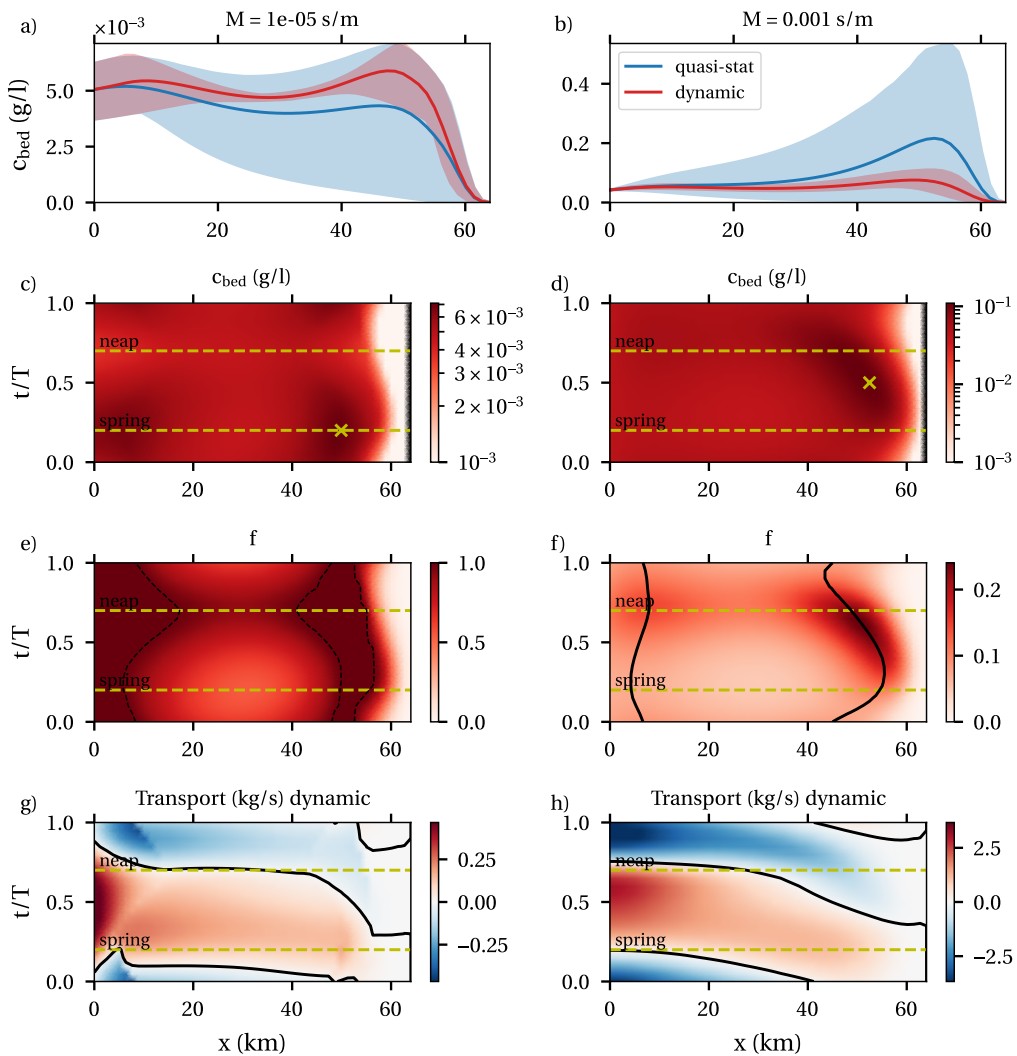

**Figure 6.** Results for model runs with low and high values of the erosion parameter $M$. a-b) mean near-bed concentrations (lines) and the range of tidally-averaged near-bed concentration (shaded). c-d) near-bed concentrations. The x marks the maximum concentration..e-f) erodibility. Black dashed lines indicate erosion limited areas. Black solid lines indicate local maxima (with $f < 1$). g-h) sediment transport.

## 3.5 Application to the hyperturbid Loire

Next, we discuss the application of the model to the Loire configuration (Dijkstra and De Goede, Submitted to Journal of Geophysical Research: Oceans), in which the model includes various additional physical processes (see Sect. 2.6.2). The additional processes mean that the sediment concentration now influences the flow (through its effect on turbulence), sediment

capacity, and transport capacity. The dynamic effects of the spring-neap cycle are therefore not only visible in the erodibility and sediment concentration, but also in the sediment capacity and transport capacity.

Fig. 7a shows near-bed average concentration (lines) and variability over the spring-neap cycle (shaded areas) in the quasi-stationary and dynamic cases. The ETM are located around km 8 and 30 with mean concentrations around 30 g/l. Fig.7b-c show the time dependence of the near-bed concentration. The ETM only move a few km over the spring-neap cycle for both the quasi-stationary and dynamic cases. In the quasi-stationary case there is a small time interval after neap tide when the ETM disappears from the estuary. In the dynamic case, however, the ETM mostly remains present during the spring-neap cycle. Exception is a small time interval near km 8 when stratification briefly collapses leading to somewhat lower concentrations. The along-channel maximum near-bed concentrations vary between 26 g/l and 52 g/l over the spring-neap cycle, with higher near-bed concentrations at neap. Surface concentrations (Fig.7d-e) have along-channel maxima which vary between 1 and 3 g/l, with the maximum occurring after spring tide.

To better understand these results, we will again apply our analysis framework. Fig. 8 shows the transport capacity scaled by the local near-bed velocity amplitude in the dynamic case at neap and spring tide. First concentrating on the total transport capacity, the downward zero-crossings, indicating the locations of maximum erodibility, are located at approximately the same locations. This helps to explain the quite minor qualitative differences between spring and neap in the erodibility and sediment concentration noted earlier. The transport capacity at spring is larger than at neap, even relative to the near-bed velocity. During spring tide, the eddy viscosity is bigger mainly due to bigger shear stresses and the effective settling velocity is smaller due to hindered settling. Hence, sediments are mixed up higher in the water column and the water column can accommodate and transport more sediment. This effect of mixing causes all contributions to the transport capacity to increase at spring. In the km 8 ETM, the baroclinic contribution is most important, closely followed by the effect of the externally generated $D_4$ tide. In the km 40 ETM, the external tidal asymmetry is dominant at neap, while spatial settling lag is dominant at spring.

The various contributions to the transport capacity do no longer scale with the tidal amplitude exactly as in Table 2 due to the added physical processes. Table 3 presents the relative differences between spring and neap magnitude of the various transport contributions (not scaled by the near-bed velocity) in the areas 0-10 km (1st column) and 20-40 km (2nd column). The relative spring-neap differences are much larger than the scaling of Table 2 (third column) suggests, especially in the first 10 km. Relatively, however, we still notice that the spatial settling lag and internal asymmetry are much more strongly amplified at spring due to their greater sensitivity to the tidal amplitude.

Next looking at the bottom pool dynamics, it is observed that the erodibility (Fig. 7f-g) is largest around the locations of the ETM, consistent with the convergence of transport capacity. The erodibility is highest around neap, with erosion limited conditions in km 0-8. The amount of sediment in the bottom pool remains of the same order of magnitude during spring and neap (not shown). Erodibility is nevertheless highest during neap because the sediment capacity is low due to small bed shear stress and strong stratification. The quasi-stationary and dynamic cases show quite similar results, except for some time after neap tide, where the quasi-stationary case shows flushing of the bottom pool. In the dynamic case, the erodibility remains significantly larger than zero. Sediment is transported (Fig. 7i) to the bottom pool between km 0-10 around neap tide ($\sim$ 0.5$<t/T<$1), and to the bottom pool at km 40 around spring tide ($\sim$ 0$<t/T<$0.5). Hence, even though trapping occurs at

approximately the same locations during the entire spring-neap cycle, sediment is transported between the two bottom pools between spring and neap.

Finally considering the SPM concentration in the dynamic case, the location of high near-bed concentrations (Fig. 7c) does not move much due to the stable locations of the bottom pools. Highest near-bed concentrations occur at neap because the erodibility is high and strong stratification keep sediment confined close to the bed. Surface concentrations (Fig. 7e) correspondingly are lowest during neap. Maximum surface concentrations occur some time after spring tide, due to larger mixing.

Summarising, we find that the transport capacity indicates similar trapping locations over the entire spring-neap cycle, with sediment transported between the two trapping areas. A combination between high erodibility and low bottom shear stress at neap and lower erodibility but higher bed shear stress at spring means quite high concentrations can be attained during the entire cycle. Observed peaks in bottom and surface concentration are mainly related to the stratification, determining whether sediment can be mixed or is confined to the bed. The differences between the dynamic and quasi-stationary cases are quite small, except for a window of a few days after neap, indicating a rapid adjustment of the estuary .This is supported by the (diagnostic) adjustment scale (Sect 3.4), which in this case is approximately 4 days.

| Contribution | $T_{\text{spring}}/T_{\text{neap}}$ 0-10 km | $T_{\text{spring}}/T_{\text{neap}}$ 20-40 km | Scaling with $|\zeta_{\text{spring}}|/|\zeta_{\text{neap}}|$ from Table 2 |
|---|---|---|---|
| River | 25-50 | 5 | $|\zeta_{\text{spring}}|/|\zeta_{\text{neap}}| = 2.17$ |
| Spatial settling lag | 100-500 | 10-20 | $(|\zeta_{\text{spring}}|/|\zeta_{\text{neap}}|)^3 = 10.3$ |
| Internal asymmetry | 100-400 | 4-10 | $(|\zeta_{\text{spring}}|/|\zeta_{\text{neap}}|)^3 = 10.3$ |
| External asymmetry | 20-30 | 4-7 | $|\zeta_{\text{spring}}|/|\zeta_{\text{neap}}| = 2.17$ |
| Baroclinic | 25-50 | 2 | $|\zeta_{\text{spring}}|/|\zeta_{\text{neap}}| = 2.17$ |
| Diffusion | 10-20 | 2 | $|\zeta_{\text{spring}}|/|\zeta_{\text{neap}}| = 2.17$ |

**Table 3.** Relative difference between the transport capacity contributions at spring and neap tide in two areas of the model domain.

## 4 Discussion

### 4.1 Wider applicability and limitations of the methodology and results

Our framework of analysis of the dynamics of SPM extends beyond the scope of this study. Firstly, such analysis may be extended to any variation in the forcing parameters that is much slower than the $D_2$ tidal timescale, such as seasonal discharge variations (see e.g. Brouwer et al, 2018). The analysis framework can partly be used to interpret results from numerical simulation models. Inclusion of spatially varying viscosity and diffusivity through more complex turbulence closures does not affect the interpretation framework. Nonlinear interactions between water motion, salinity, SPM and mixing within such turbulence closures do complicate a cause-and-effect analysis in the same way as we found in the Loire case, but otherwise still allows use of the analysis framework. Furthermore, it is possible to compute the bottom pool thickness, erodibility and total sediment

transport capacity from such models (see Dijkstra, 2019, Ch 7 for an example in the Delft3D model). Decompositions of the transport capacity, such as the ones used in this study, can usually not be computed from simulation models.

The aim of this study was to provide insights into various effects of the spring-neap cycle on SPM dynamics, but ignored various potentially important aspects. Some aspects that were not considered in the idealised case, such as the influence of velocity, salinity and sediment concentration on turbulence parameters, were added in the Loire case study. Still, $D_2$ tidal variations in the salinity and turbulence parameters were not considered, thus excluding effects of tidal straining and eddy viscosity-shear covariance (ESCO). These processes would affect the trapping and resuspension dynamics in a complex way.

Some other aspects that were ignored are worth mentioning specifically for further study. Firstly, the width averaging means the lateral dynamics is ignored. In estuaries it is well possible that sediment accumulates on shallow flanks with erosion limited conditions, while supply limited conditions dominate in the deeper channel. The effects of spring-neap variations in such contexts still needs further investigation. Also, it is assumed that all sediment in the bottom pool remains easily erodible without a critical threshold for the bed shear stress. Consolidation of sediments on the spring-neap timescale is not taken

into account and may additionally significantly affect the dynamics. Thirdly, the present model only considers changes in the bottom pool over the spring-neap cycle as a consequence of the tidally averaged transport. This ignores the intratidal variations of the bottom pool thickness.

The two-timescales formalism as applied here ignores some dynamics that is assumed to be unimportant. For example, it was assessed that the interaction between the tidal and spring-neap cycle in the sediment balance is negligible, but this assumption

still needs a-posteriori verification. Also, it was assumed that the water motion adapts quickly to spring-neap conditions. Allen et al (1980, (p.75)) argue that overall storage of water in the estuary can potentially be larger during spring tide than neap tide due to a different subtidal water set-up. Hence there is a subtidal flow of water in and out of the estuary over the spring-neap cycle. Our scaling arguments indicate this effect to be negligible, but to the author's knowledge, this has not been systematically verified.

**4.2  Use of quasi-stationary model simulations**

Quasi-stationary model results are frequently used to gain understanding of estuarine sediment dynamics (e.g. Burchard and Baumert, 1998; Chernetsky et al, 2010; McSweeney et al, 2016; Dijkstra et al, 2019). Since quasi-stationary results do not depend on the forcing history but are consistent with the actual (tidal) forcing, this greatly reduces complexity. In cases where the dynamics of e.g. the spring-neap cycle is important, some caution needs to be taken in the interpretation of such results. First

considering the case where quasi-stationary results are used to study spring-neap averaged results, it is clear from our results that spring-neap mean sediment concentrations in the dynamic and quasi-stationary cases are not the same. By calibration using e.g. the erosion parameter, similar concentrations between the dynamic mean and quasi-stationary case may be nevertheless obtained and hence one should be careful that the value of such a calibration parameter is highly context dependent. As the contributions to the sediment transport capacity scale non-linearly with the tidal amplitude and $D_2$-$D_4$ phase difference, using

an average spring-neap amplitude and phase difference results in an error in the relative importance of processes, which may result in a shift of the sediment trapping location. As long as similar processes are important during spring and neap tide, the

dominant processes are still identified by taking a spring-neap average. Therefore, the quasi-stationary simulations should be more regarded as qualitative than quantitatively correct when dynamics are important.

Secondly, we consider the use of quasi-stationary reasoning to understand the dynamics during spring or neap tidal phases.
From our results it is concluded this should be done with caution, since the dynamic sediment concentrations may be quite different from the quasi-stationary ones, as there are significant memory effects. Hence, one should best use these results to indicate a range of the dynamics occurring over the spring-neap cycle, acknowledging that the dynamic spring-neap variability in SPM concentration is smaller.

## 5 Conclusions

In this study we set-up and applied a framework for understanding spring-neap dynamics of fine sediments in estuaries. The framework consists of analysing sediment trapping using sediment transport capacity, consequently considering the slow movement of the bottom pool and finally studying the local interaction between the bottom pool and water column to understand the SPM concentrations. Our framework is motivated using a two-timescales approach, which separates the solution over the tidal timescale and the spring-neap timescale. It shows that it is mainly the slow movement of the bottom pool that explains
lag or memory effects, while the water motion and vertical distributions of SPM adapt much faster. The use of our framework was demonstrated using an idealised test case where the sediment dynamics does not affect the water motion, and a case representing the Loire estuary, with significant feedback between the SPM concentration and water motion.

In the idealised test case it was shown that dynamic effects of the spring-neap cycle are quite important. While transport capacity could indicate flushing of the ETM at some stage of the spring-neap cycle, slow adaptation of the bottom pool can
cause the ETM to persist over the entire spring-neap cycle. Depending on the interplay between this dynamically influenced bottom pool, the bed shear stress and stratification, the timing of the maximum concentrations during the spring-neap cycle can be quite different. This was illustrated by comparing the same test case with different values of the erosion parameter, showing maximum SPM concentrations at opposite stages of the spring-neap cycle. The framework also proved useful interpreting the results of the Loire case. Due to the feedback, cause-and-effect between water motion, transport and SPM concentration could
not be considered, but we could still explain the trapping locations and timing of maximum concentrations in a systematic way.

*Code availability.* The model code and input files to reproduce the model experiments are available under iFlow version 3.1 on GitHub (https://github.com/iFlow-Modelling-Framework/iFlow, or https://doi.org/10.5281/zenodo.822394).

## Appendix A: Mathematical computation of quasi-stationary conditions

The quasi-stationary results used in this study are computed by solving the model equations presented in Section 2.1 with
520 some minor modification. In quasi-stationary conditions, one can identify two different cases. In the first, the sediment stock $S$ is in balance with the instantaneous hydrodynamics conditions, which means that slow-time derivative of the stock vanishes

($S_{t_2} = 0$ in Eq. (12)). In the second case, the sediment concentration is in balance with the instantaneous hydrodynamic conditions, but the bottom pool keeps growing. The growing bottom pool can however not be fully re-suspended and does not alter sediment concentration. It is assumed that the bottom pool remains small enough not to affect the water depth. In both cases, the slow-time derivative of the erodibility vanishes. To use this latter observation, we rewrite Eq. (12). Note that erodibility $f$ is a function of the stock $S$. Multiplying Eq. (12) by $\frac{\partial f}{\partial S}$ and using that $\frac{\partial f}{\partial S} S_{t_2} = f_{t_2}$ we obtain:

$$B\delta f_{t_2} = -\frac{\partial f}{\partial S} \left\langle B \int_{-H}^{\zeta} uc - K_h c_x \, dz \right\rangle_x . \tag{A1}$$

Assuming quasi-stationary conditions we set $f_{t_2} = 0$ and solve the above equation together with the equations in Section 2.1.

On a technical note, $\frac{\partial f}{\partial S}$ may be equal to zero. Hence, multiplying Eq. (12) by $\frac{\partial f}{\partial S}$ eliminates that equation if $\frac{\partial f}{\partial S} = 0$. This is not a problem however, since $\frac{\partial f}{\partial S} = 0$ only happens when $f = 1$, so we also eliminate one of the unknowns of the model. The model hence remains solvable.

### Appendix B: Additional information for the Loire case

Table B1 lists the parameter values used in the Loire case. Please see Dijkstra and De Goede (Submitted to Journal of Geophysical Research: Oceans) for a further description of the model.

| | Parameter | Value | | Parameter | Value |
|---|---|---|---|---|---|
| $A_{M_2}$ | $M_2$ amplitude at $x = 0$ | 1.75 m | $A_{M_4}$ | $M_4$ amplitude at $x = 0$ | 0.20 m |
| $A_{S_2}$ | $S_2$ amplitude at $x = 0$ | 0.65 m | $A_{S_4}$ | $S_4$ amplitude at $x = 0$ | 0.007 m |
| $\phi_{M_2}$ | $M_2$ phase at $x = 0$ | 0 deg | $\phi_{M_4}$ | $M_4$ phase at $x = 0$ | -148 deg |
| $\phi_{S_2}$ | $S_2$ phase at $x = 0$ | 103 deg | $\phi_{S_4}$ | $S_4$ phase at $x = 0$ | 110 deg |
| $Q$ | River discharge | 250 m²/s | $w_{s,0}$ | Clear water sediment settling velocity | 2 mm/s |
| $z_{0,\text{est}}^*$ | Dimensionless roughness estuary ($x < 45$ km) | 0.01 | $c_{\text{sea}}$ | Depth-averaged tide-averaged sediment concentration at $x = 0$ | 0.3 g/l |
| $z_{0,\text{riv}}^*$ | Dimensionless roughness river ($x > 60$ km) | 0.05 | $K_h$ | Horizontal eddy diffusivity | 100 m²/ |
| $M$ | Erosion parameter | 0.03 s/m | $c_{\text{gel}}$ | Gelling concentration | 100 g/l |

**Table B1.** Default values of the parameters in the Loire case.

The subtidal eddy viscosity and eddy diffusivity in this case depends on the velocity and stratification through the equations

$$A_\nu = \left\langle c_{\nu,1}(z_0^*)|U|(H+\zeta)F(\overline{\text{Ri}}) \right\rangle, \tag{B1}$$

$$K_\nu = \left\langle \frac{c_{\nu,1}(z_0^*)}{\sigma_\rho}|U|(H+\zeta)G(\overline{\text{Ri}}) \right\rangle. \tag{B2}$$

Here, $c_{\nu,1}(z_0^*)$ is a coefficient that depends on a calibrated dimensionless roughness height $z_0^*$, $\sigma_\rho$ is a constant Prandtl-Schmidt number of 1, $|U|$ is the depth-averaged velocity magnitude, and $F(\overline{\mathrm{Ri}})$, $G(\overline{\mathrm{Ri}})$ are adaptations of the Munk and Anderson (1948) functions for stratification-induced damping of turbulence: $F(\overline{\mathrm{Ri}}) = \left(1 + 10\overline{Ri}\right)^{-1/2}$ and $G(\overline{\mathrm{Ri}}) = \left(1 + 3.33\overline{Ri}\right)^{-3/2}$. They depend on the depth-average $(\bar{\cdot})$ of an approximation of the Richardson number

$$\mathrm{Ri} == -g\frac{\beta_c c_z + \beta s_z}{u_z^2 + u_{z,\mathrm{min}}^2}. \tag{B3}$$

This is best thought of as a bulk-Richardson number. Here $u_{z,\mathrm{min}} = 0.03\ \mathrm{s}^{-1}$ parametrises the unresolved shear, e.g. by lateral flows and $\beta_c = 6.2 \cdot 10^{-4}\ \mathrm{m}^3/\mathrm{kg}$.

The resulting eddy viscosity, Richardson number and the depth-averaged $M_2$ velocity magnitude are plotted in Fig. B1.

*Author contributions.* Conceptualization: YMD,DDB,HMS. Investigation: YMD, DDB. Software: YMD, DDB. Methodology: YMD,DDB,HMS. Visualization: YMD. Writing - original draft: YMD. Writing - review: HMS.

*Competing interests.* The authors declare they have no competing interest.

*Acknowledgements.*

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

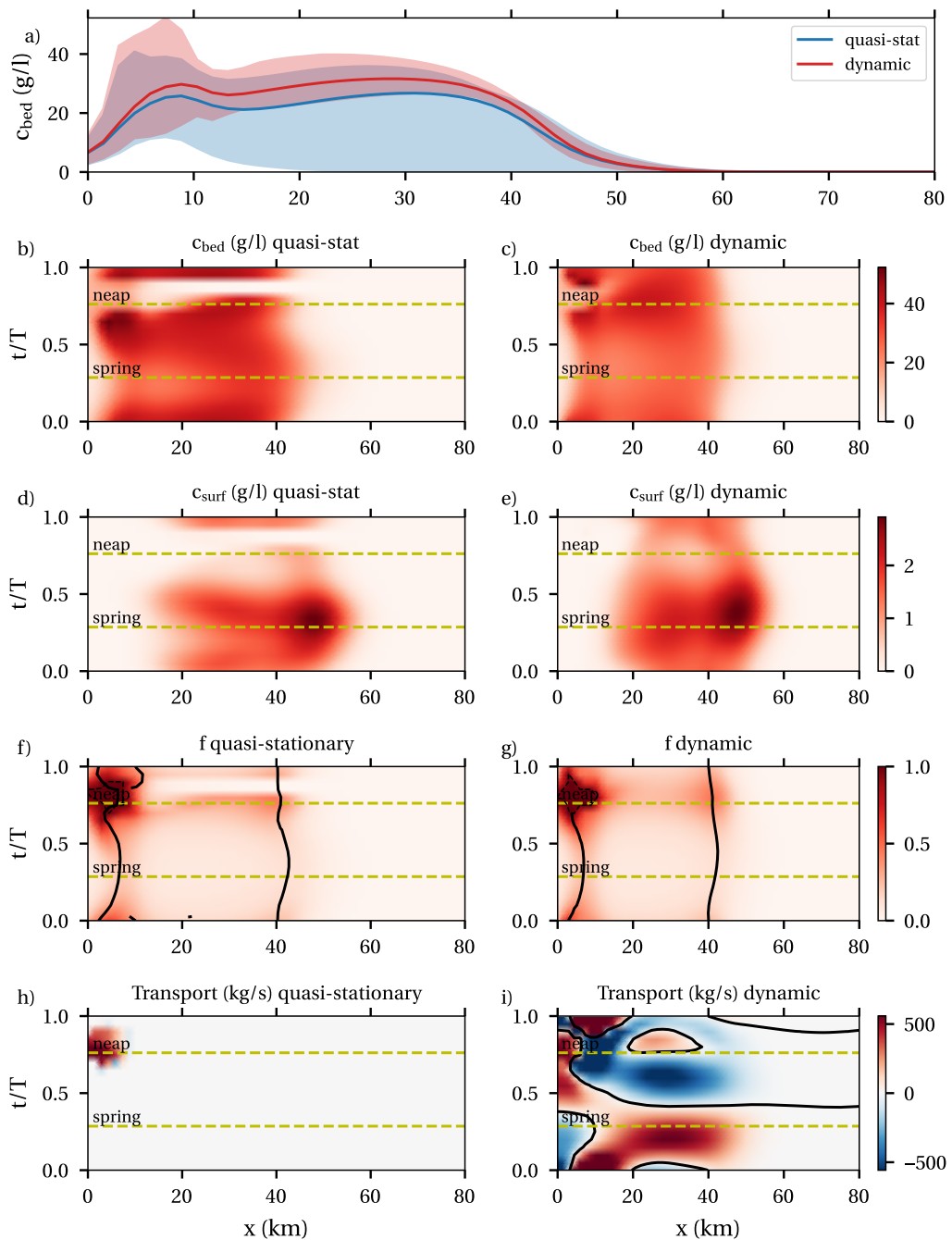

**Figure 7.** Results for the sediment concentration in the Loire case, comparing quasi-stationary and dynamic results. a) mean near-bed concentration (lines) and the spring-neap range of the tidally-averaged near bed concentration (shaded area). b-c) near-bed concentrations in the quasi-static and dynamic cases as function of $x$ and $t$. d-e) id. for surface concentrations. f-g) erodibility. Black dashed lines indicate erosion limited areas. Black solid lines indicate local maxima (with $f < 1$). h-i) sediment transport.

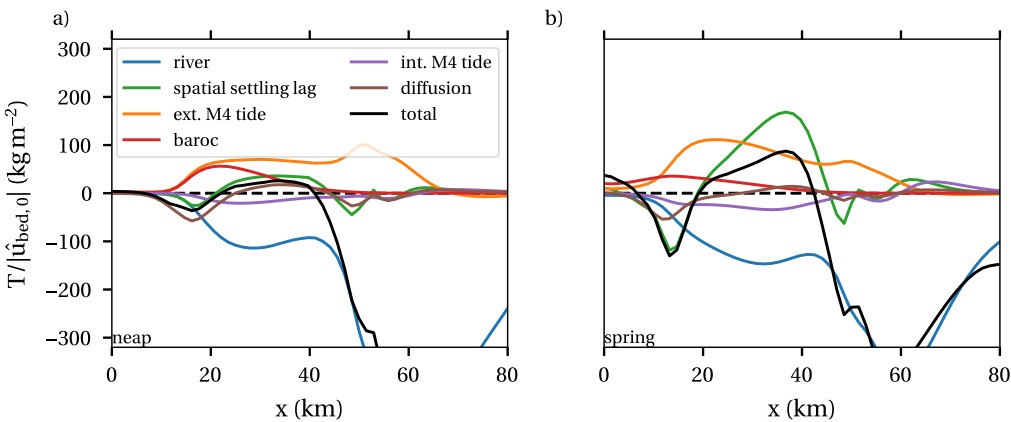

**Figure 8.** Transport capacity (black) and the leading contributions to the transport capacity for neap (a) and spring (b) for the Loire case.

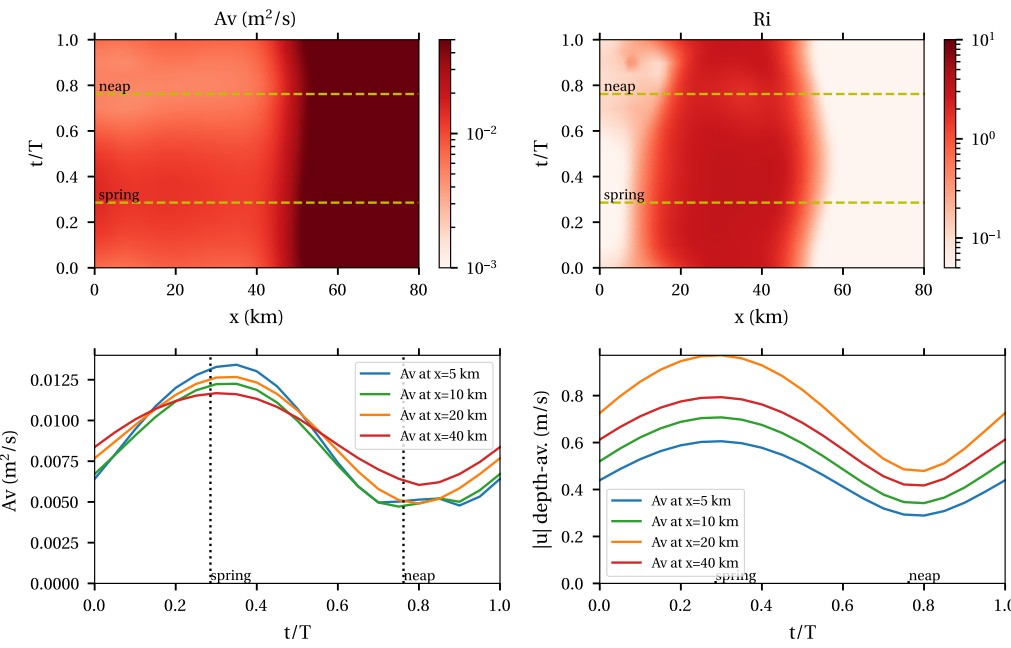

**Figure B1.** Right: eddy viscosity in time relative to the spring-neap period ($T$) and space ($x$) in the Loire case. Left: the depth-averaged (bulk) Richardson number (top) and depth-averaged $M_2$ velocity magnitude (bottom).