# Peer review of "Disentangling spring-neap SPM dynamics in estuaries"

_EGUsphere, 2024_

## Referee Comment (RC1)

Review on 'egusphere-2024-1364: Disentangling spring-neap SPM dynamics in estuaries'.

The article investigates the variations in suspended particulate matter (SPM) concentrations in estuaries, particularly in relation to the spring-neap tidal cycle. It introduces a structured framework designed to analyze the factors contributing to these variations, emphasizing the roles of sediment transport capacity and bottom pool dynamics. This framework aims to enhance the understanding of the complex interactions between sediment trapping, re-suspension, and transport within estuarine environments. The study presents two model cases implemented in the iFlow model to demonstrate the utility of this framework: an idealized test case and a case representative of the Loire estuary.

While the article provides a detailed examination of SPM dynamics and proposes a useful method for systematic analysis, there are several points that could benefit from further exploration and clarification. E.g. how the bed shear stress and stratification combine to determine the magnitude of mixing, therefore influence the ETM? What are their relative roles? My specific comments are as follows.

1. Near line 15: The full name of ETM (Estuarine Turbidity Maximum) should be given when it first appears.

2. Line 30: what is the contribution of baroclinic tides (internal tides) to the mixing here? is the mixing mainly induced by internal tidal shear?

3. Line 35: Is there a positive feedback between tidal mixing and shear stress? How do shear stress and tidal mixing individually and collectively contribute to sediment dynamics?

4. Line 40: What is the full name of SSC? I believe this is the first time this abbreviation appears.

5. Line 40: Can you refer to it as an interaction, or is it merely a relation?

6. Line 65: Perhaps you should provide a schematic figure illustrating the model domain and settings. This would be much more informative.

7. Line 80: In the real ocean, these two parameters are highly likely to be inhomogeneous. You may want to discuss this point and the challenges associated with using a more realistic parameterization.

8. Line115: How do you specify the M4 component? The nonlinear effects of M2 will also generate M4 frequencies.

9. Line 150: typo 'columnis' and 'i..e.'

10. Line 205: what would be the contribution of baroclinic tide, comparing to low-frequency baroclinic currents?

11. Line 240: Can you explain the parameters are set to which fixed value and why?

12. Line 245: what is the vertical resolution for the model?

13. Line 270: The ETM is not very distinct near the 8 km mark. What causes the rapid growth and decrease of the ETM before and after the 55 km point?

14. Line 285: I also observe a transition point near the 17 km mark during the spring tide.

15. Line 300: Most of the explanations here focus on the mathematical perspective. I believe a more dynamic explanation is needed. E.g. What is the dynamical difference between internal and external D4 forcing?

16. Line 325: This somewhat contrasts with the aforementioned statement. The strong bed shear stress definitely plays an essential role here. How can you argue that the bottom pool is more important? Can you explain more in detail?

17. Line 395: In Figure 6c, I can also identify a time period with an abruptly decreased ETM at 8 km point, so saying 'ETM remains present' might not be accurate. Perhaps some discussion on why this interval appears near the 8 km mark but is not as significant as in the quasi-static case is needed.

18. Line 400: The mixing magnitude is very interesting and worthy of exploration. Could you show the relative roles of shear stress and stratification in determining the magnitude of mixing here? Is the Osborn relation valid for estimating the mixing magnitude in this context?

19. Line 415: Can you also show the variations in shear stress and stratification, as well as the corresponding variations in mixing?

---

## Referee Comment (RC2)

Review of the manuscript No.: egusphere-2024-136

Title: "Disentangling spring-neap SPM dynamics in estuaries"

Authors: Yoeri M. Dijkstra, Dennis D. Bouwman, and Henk M. Schuttelaars

**Preliminary remark.** I collaborate closely with two authors (YMD and HMS), though not on this particular topic. Nonetheless my comments would make this probably clear. Hence I have unveiled my identity to the authors. Any discussion on this paper, however, has been and will be in writing only or will at least be made reproducible in some way.

**General comments.**

The authors extend an existing numerical tool (Brouwer et al. 2018) that describes long-term subtidal variations by discharge to include spring-neap behavior as well. This model is applied to two cases. First the authors consider an idealized estuary that is somewhat inspired on the Ems. In this case the backreaction of sediment on water motion ("turbulent damping") and non-linear effects like hindered settling are absent. The second case specifically describes the hyperturbid Loire estuary and does include turbulent damping and hindered settling.

Three (although maybe four) analysis tools are defined and adopted to enhance interpretation of the model results, concentrating on horizontal and vertical sediment dynamics as well identification of dominant processes that govern the horizontal sediment balance.

The model setup and solution method (Sect. 2.1-2.3) are well written. I also applaud the approach to identify the role of individual processes, as this is an inherent advantage of the adopted model approach.

However, I do think that a number of things need to be clarified. Among these are the nature of the quasi-stationary runs, in general the analysis methods as outlines in Sect. 2.4 and the way they are adopted when interpreting the model runs. Below, I will elaborate on these points.

I find this paper acceptable with minor revisions provided that these concerns are addressed.

**Nature of the quasi-stationary runs.**

The authors state (lines 190-191) that the quasi-static runs are done by setting the time derivative of stock S to zero in Eqs. (11)-(12). I was very confused by this as this precludes the possibility of mudpools being formed which do occur in the quasi-stationary state. Moreover from the initial condition S=0 (Sect. 2.3) no sediment stock is expected to form at all. This cannot be correct.

I have given it some thought and it seems to me that the authors actually use the instantaneous D2 and D4 forcing at each time step to compute the corresponding equilibrium sediment distribution (which indeed may contain erosion limited regions, hence mud pools).

To me, further explanation of what the quasi-stationary runs are and how they can lead to mud pool formation is crucial to understand the authors' findings. As it is presented now, all

discussion of 'mud pools', 'erosion limitation' and 'f=1' with reference to the quasi-stationary runs seem inconsistent to me.

**Analysis methods (Sect. 2.4) and their applications (Sects. 3.2 and 3.5)**

Here I lost count of the actual number of tools the authors use: they claim three (first on line 185, second on line 210 and third on line 220). However, the D2-tide averaged results (which give the horizontal transport capacity and individual physical mechanisms) on lines 192-209 seem to be a fourth one, which is now confusingly described under 'Next, .. '.

I would propose that the authors to make clear whether there are 3 of 4 analysis tools and (in the former case) motivate why lines 185-209 are one tool. Use additional section numbering (2.4.1 to 2.4.3 / 2.4.4) to introduce each analysis tool in a slightly more structural way. I would also recommend to have their analysis sections (3.2 and 3.5) discussed in a similar way (i.e. as subsubsections). Sections 3.2 and 3.5 really contain a lot of information, linking the analysis explicitly to the tools (subsections) in Sect. 2.4 would be most helpful.

I think the authors may have to slightly rewrite the abstract as well to keep consistent with Sect. 2.4. For instance, sediment capacity is mention as an analysis tool while this is not mentioned as such in Sect. 2.4. Indeed, in Sect. 2.4 a comparison between quasi-stationary and dynamic runs is presented as an analysis tool (but not in the abstract).

I think this will provide a more convenient guide for the reader who is often not familiar with the authors' approach and has to absorb quite an amount of information along the way.

**Further remarks**

1. Line 20: "This leads" → "This *may* lead". To me trapping, as characterized by e.g. ETMs, does not necessarily imply erosion limited conditions.
2. Line 77: "in over" → "over" (typo)
3. Line 195-196: "were added on the bed", perhaps extend to "were added on the bed, i.e. global erosion limited conditions (f=1)." To me, this would clarify the definition od sediment transport capacity as being the maximum sediment transport that is possible under local hydrodynamic conditions.
4. Line 262-263: "For alle model settings ... refer to DgD24". I would give the authors the consideration to include the settings in an Appendix. After all, DGD24 has not yet been accepted yet...
5. Line 400-402: The downward zero-crossing at 8 km is not well visible during neap tide.
6. Line 402-403: "This helps to explain the quite minor differences between spring and neap ... noted earlier.". I think this should be minor *qualitative* differences as the magnitude of the concentration varies by a factor two.
7. Line 403-405: To what extend can hindered settling also contribute to sediments being kept higher in the water column and thus contribute to the increased sediment capacity?
8. Line 418-419: "Erodibility is nevertheless ... and strong stratification." To me, this may be seen as mimicking a situation of low erosion parameter M which indeed corresponds to a greater likelihood of erosion limitation (see Fig. 5).
9. Line 420-421: "..., where the quasi-stationary case shows flushing of the bottom pool. In the dynamic case, the bottom pool remains present". I agree that the bottom pool is

flushed in the quasi-stationary case. However, as it it written here I interpret the remark about the dynamic case as that the bottom pool is *always* present which is not correct. Indeed, f=1 occurs only after neap at the downstream located ETM. Please explain that there is a permanent mud pool or restate this remark.

10. Line 420-423: I agree that sediment is transported to the bottom pool at the entrance at neap. I also agree that at spring there is transport towards the ETMs at 40 km, but I don't think there is a bottom pool there (f<1 at spring, see previous remark). I think that 'trapping areas' is more appropriate here. Besides, it also seems that there is accumulation of sediment at the 40 km ETM after neap.

11. Line 423: "..., sediment is transported between the two bottom pools between spring and neap". First, I only think there is one bottom pool (see previous remarks above). Second, I found this not so clear from what the authors wrote. I think that sediment is being transported from the downstream to the upstream ETM at 0.5<t/T<neap (blue region) and visa versa for 0<t/T<0.35 (red region). Is this what the authors want to convey? I would think that this back and forth transport is necessarily a recirculation (both net transports being equal) since the authors consider a situation that is equilibrium on the neap-spring timescale. Could the authors comment on this?

---

## Author Comment (AC1)

Dear editor, dear reviewers,

We thank the reviewers for their detailed assessment of our manuscript and their helpful suggestions. We have included our response to both reviewers in red below.

Kind regards,

Yoeri Dijkstra, Dennis Bouwman and Henk Schuttelaars

**Reviewer #1**

Review on 'egusphere-2024-1364: Disentangling spring-neap SPM dynamics in estuaries'.

The article investigates the variations in suspended particulate matter (SPM) concentrations in estuaries, particularly in relation to the spring-neap tidal cycle. It introduces a structured framework designed to analyze the factors contributing to these variations, emphasizing the roles of sediment transport capacity and bottom pool dynamics. This framework aims to enhance the understanding of the complex interactions between sediment trapping, re-suspension, and transport within estuarine environments. The study presents two model cases implemented in the iFlow model to demonstrate the utility of this framework: an idealized test case and a case representative of the Loire estuary.

While the article provides a detailed examination of SPM dynamics and proposes a useful method for systematic analysis, there are several points that could benefit from further exploration and clarification. E.g. how the bed shear stress and stratification combine to determine the magnitude of mixing, therefore influence the ETM? What are their relative roles? My specific comments are as follows.

1.      Near line 15: The full name of ETM (Estuarine Turbidity Maximum) should be given when it first appears.

Here changed to SPM. ETM is introduced later.

2.      Line 30: what is the contribution of baroclinic tides (internal tides) to the mixing here? is the mixing mainly induced by internal tidal shear?

In our literature review here we do not focus on the role of the internal tides, but rather on the subtidal effects. Mixing is then, at least conceptually, regarded as the result of tidal velocity (bed shear stress and velocity shear in the water column) and subtidal stratification. Literature, e.g. the mentioned study of Jay and Musiak, and further e.g. Burchard & Baumert (1998) address the role tidal straining and tidal mixing variations on sediment trapping. Since our model does not use a tidally varying eddy viscosity and ignores the effect of tidally varying salinity, we chose here not to consider internal tides in the literature review. In reply to another comment below, we have added a brief discussion on not including effects of tidal straining in Section 4.1.

3.      Line 35: Is there a positive feedback between tidal mixing and shear stress? How do shear stress and tidal mixing individually and collectively contribute to sediment dynamics?

We assume the reviewer means to ask how this works in our model. In our idealised case study, the tidal mixing is parametrised by a constant eddy viscosity which does not depend on shear stress or tidal velocity. In the Loire case, the eddy viscosity does depend on the tidal velocity amplitude, which is a proxy for the shear stress.

Although the constant eddy viscosity in the idealised test case is not very realistic, we found it useful to separate the effect of changing shear stress from changes in mixing or trapping processes as much as possible. Results are extensively discussed in the main text. The collective effect of shear stress and mixing are discussed in context of the Loire case. Due to the nonlinearity, it is hard to separate the effects. This can be done by switching of feedback between water motion and tidal mixing (i.e. choosing fixed Av), but we believe this is somewhat beyond the scope of this work.

4.      Line 40: What is the full name of SSC? I believe this is the first time this abbreviation appears.

Suspended sediment concentration; has now been added.

5.      Line 40: Can you refer to it as an interaction, or is it merely a relation?

In general one could say all processes interact. If sediment concentrations are low, one could say the hydrodynamics processes interact yielding a relation to the SSC. If SSC is high, hydrodynamics and sediment dynamics interact as well. We believe the text is sufficiently clear at this line, and this question applied to our model is further addressed in the set-up of the two cases.

6.      Line 65: Perhaps you should provide a schematic figure illustrating the model domain and settings. This would be much more informative.

Done

7.      Line 80: In the real ocean, these two parameters are highly likely to be inhomogeneous. You may want to discuss this point and the challenges associated with using a more realistic parameterization.

It is not quite clear which two parameters the reviewer refers to, since the line features ws, Kv and Kh, so we consider all three. Horizontal variations of ws and Kv are included in the Loire set-up, while Kh is always assumed constant. Vertical variations are not considered in this study, so that the vertical profiles remain fairly simple (and analytical), which speeds up the model. However, the interpretation framework is applicable without any additional considerations when nonuniform parameters would be included. This is explicitly added to discussion section 4.1.

8.      Line115: How do you specify the M4 component? The nonlinear effects of M2 will also generate M4 frequencies.

The M4 (or D4) is only prescribed at the seaward boundary. Generation of M4 (or D4) due to M2 (or D2) inside of the estuary is explicitly resolved by the model.

9.      Line 150: typo 'columnis' and 'i..e.'

Changed

10.     Line 205: what would be the contribution of baroclinic tide, comparing to low-frequency baroclinic currents?

This was not investigated. The model does not resolve the tidal variations of salinity and hence the baroclinic tides. Adding tidal variations in mixing and salinity could alter both the resuspension and trapping. The effect is quite complex since it affects exchange flow (ESCO circulation) and mixing. It is not our aim to fully include all processes in this study, but rather to

illustrate a way of analysing spring-neap variations in SPM dynamics. A remark on not including tidal variations in mixing and salinity was added to Section 4.1.

11.     Line 240: Can you explain the parameters are set to which fixed value and why?

The salinity profile is chosen to roughly match the profile typically found in the Ems estuary (stated later in the text). The parameters are fixed to simplify the analysis. This was added to the text. Salinity does vary with the spring-neap cycle in the Loire case.

12.     Line 245: what is the vertical resolution for the model?

All vertical profiles are resolved analytically. This was added to the text.

13.     Line 270: The ETM is not very distinct near the 8 km mark. What causes the rapid growth and decrease of the ETM before and after the 55 km point?

This is explained in Fig 4: there is a rapid decrease in the importing capacity, especially the internally generated M4 tide. After the 55 km point, the tidal velocity decreases rapidly towards the weir at the landward end, so that both the importing capacity as well as the bottom shear stress decrease rapidly.

14.     Line 285: I also observe a transition point near the 17 km mark during the spring tide.

We believe the reviewer means the zero crossing from negative to positive in fig 4b. This is a divergence point where one expects a minimum in the bottom pool thickness. We only consider convergence points.

15.     Line 300: Most of the explanations here focus on the mathematical perspective. I believe a more dynamic explanation is needed. E.g. What is the dynamical difference between internal and external D4 forcing?

We have added the explanation that the external $D_4$ transport relates to the tidal asymmetry between the linear parts of the D2 and D4 tides. The internal D4 tide is caused by quadratic interactions of the D2 tide due to processes including advection and stokes drift. The transport related to this is the asymmetry of the D4 tide and the D2 tide and hence scales with the D2 amplitude cubed.

16.     Line 325: This somewhat contrasts with the aforementioned statement. The strong bed shear stress definitely plays an essential role here. How can you argue that the bottom pool is more important? Can you explain more in detail?

After giving this some more thought we have decided to remove this sentence. While we found it appealing at first to point to one variable as the 'leading factor', your comment made us realise this is not quite right. The point of our analysis framework is that the combination of transports, (delayed) bottom pool movement and re-suspension governs the ETM dynamics, so also within this conclusion it is not quite right to emphasise one variable more than the others.

17.     Line 395: In Figure 6c, I can also identify a time period with an abruptly decreased ETM at 8 km point, so saying 'ETM remains present' might not be accurate. Perhaps some discussion on why this interval appears near the 8 km mark but is not as significant as in the quasi-static case is needed.

We have added "Exception is a small time interval near km 8 when stratification briefly collapses leading to somewhat lower concentrations." It turns out that it is quite hard to pin-point exactly

why stratification collapses here. Bed shear stresses around this time are relatively low (though not at their minimum) and it is probably some interaction between this, a slightly upstream movement of the bottom pool and the tidal asymmetry at that moment that explains this brief collapse of turbulence.

18.     Line 400: The mixing magnitude is very interesting and worthy of exploration. Could you show the relative roles of shear stress and stratification in determining the magnitude of mixing here? Is the Osborn relation valid for estimating the mixing magnitude in this context?

We have added plots of the eddy viscosity, depth-averaged tidal velocity (good proxy for the bed shear stress) and the depth-mean Richardson number (probably most closely resembling what is sometimes called the bulk Richardson number) below and in Appendix B of the manuscript, alongside the equation used to compute the eddy viscosity. We feel the figure is not needed in the main text, but nice for interested readers to find in the appendix.

The Osborn relation was not considered. Please note that the eddy viscosity is vertically uniform in our model, so it would make most sense that such a relation should be evaluated in some depth-averaged way.

19.     Line 415: Can you also show the variations in shear stress and stratification, as well as the corresponding variations in mixing?

Please see our reply above and the figure copied below. The eddy viscosity is mainly bigger during spring than neap due to changes in velocity amplitude, while the Richardson number varies but has less influence on the relative differences between spring and neap.

We have made a few changes to the main text to make it more self-sufficient to understand the changes without needing the figures of mixing.

[Figure]

**Reviewer #2 George Schramkowski**

Review of the manuscript No.: egusphere-2024-136

Title: "Disentangling spring-neap SPM dynamics in estuaries"

Authors: Yoeri M. Dijkstra, Dennis D. Bouwman, and Henk M. Schuttelaars

**Preliminary remark.** I collaborate closely with two authors (YMD and HMS), though not on this particular topic. Nonetheless my comments would make this probably clear. Hence I have unveiled my identity to the authors. Any discussion on this paper, however, has been and will be in writing only or will at least be made reproducible in some way.

**General comments.**

The authors extend an existing numerical tool (Brouwer et al. 2018) that describes long-term subtidal variations by discharge to include spring-neap behavior as well. This model is applied to two cases. First the authors consider an idealized estuary that is somewhat inspired on the Ems. In this case the backreaction of sediment on water motion ( "turbulent damping") and non-linear effects like hindered settling are absent. The second case specifically describes the hyperturbid Loire estuary and does include turbulent damping and hindered settling.

Three (although maybe four) analysis tools are defined and adopted to enhance interpretation of the model results, concentrating on horizontal and vertical sediment dynamics as well identification of dominant processes that govern the horizontal sediment balance.

The model setup and solution method (Sect. 2.1-2.3) are well written. I also applaud the approach to identify the role of individual processes, as this is an inherent advantage of the adopted model approach.

However, I do think that a number of things need to be clarified. Among these are the nature of the quasi-stationary runs, in general the analysis methods as outlines in Sect. 2.4 and the way they are adopted when interpreting the model runs. Below, I will elaborate on these points.

I find this paper acceptable with minor revisions provided that these concerns are addressed.

**Nature of the quasi-stationary runs.**

The authors state (lines 190-191) that the quasi-static runs are done by setting the time derivative of stock S to zero in Eqs. (11)-(12). I was very confused by this as this precludes the possibility of mudpools being formed which do occur in the quasi-stationary state. Moreover from the initial condition S=0 (Sect. 2.3) no sediment stock is expected to form at all. This cannot be correct.

I have given it some thought and it seems to me that the authors actually use the instantaneous D2 and D4 forcing at each time step to compute the corresponding equilibrium sediment distribution (which indeed may contain erosion limited regions, hence mud pools).

To me, further explanation of what the quasi-stationary runs are and how they can lead to mud pool formation is crucial to understand the authors' findings. As it is presented now, all

discussion of 'mud pools', 'erosion limitation' and 'f=1' with reference to the quasi-stationary runs seem inconsistent to me.

The reviewer is correct. The statement that we set S_t=0 in the quasi-stationary results is incorrect; it holds only for the areas where f<1. We have added Appendix A to explain the formal procedure for computing quasi-stationary conditions.

**Analysis methods (Sect. 2.4) and their applications (Sects. 3.2 and 3.5)**

Here I lost count of the actual number of tools the authors use: they claim three (first on line 185, second on line 210 and third on line 220). However, the D2-tide averaged results (which give the horizontal transport capacity and individual physical mechanisms) on lines 192-209 seem to be a fourth one, which is now confusingly described under 'Next, .. '.

I would propose that the authors to make clear whether there are 3 of 4 analysis tools and (in the former case) motivate why lines 185-209 are one tool. Use additional section numbering (2.4.1 to 2.4.3 / 2.4.4) to introduce each analysis tool in a slightly more structural way. I would also recommend to have their analysis sections (3.2 and 3.5) discussed in a similar way (i.e. as subsubsections). Sections 3.2 and 3.5 really contain a lot of information, linking the analysis explicitly to the tools (subsections) in Sect. 2.4 would be most helpful.

We understand the confusion. Actually, our analysis consists of three elements: analysis of transport, bottom pool and re-suspension. The comparison to quasi-stationary results is not a necessary element of the analysis framework, but we use it in the results to illustrate the importance of including the dynamics on the slow timescale. We have rewritten section 2.4 to now clearly list the three elements, moving the comparison to quasi-stationary model runs to a new section (2.5).

The same three elements form three paragraphs in section 3.2 and 3.5. We have considered the suggestion to include further subsections, but feel this would break the flow of the text too much; sections 3.2 and 3.5 are each one page long, so breaking each in three subsections leads to a lot of fragmentation.

I think the authors may have to slightly rewrite the abstract as well to keep consistent with Sect. 2.4. For instance, sediment capacity is mention as an analysis tool while this is not mentioned as such in Sect. 2.4. Indeed, in Sect. 2.4 a comparison between quasi-stationary and dynamic runs is presented as an analysis tool (but not in the abstract).

The text in the abstract was correct and the confusion was created by section 2.4. By our restructuring of section 2.4 the abstract should again be consistent with the main text.

I think this will provide a more convenient guide for the reader who is often not familiar with the authors' approach and has to absorb quite an amount of information along the way.

**Further remarks**

1.      Line 20: "This leads" → "This *may* lead". To me trapping, as characterized by e.g. ETMs, does not necessarily imply erosion limited conditions.

We do not want to associate a bottom pool with erosion limited conditions only. Also for f<1 there is sediment on the bed during part of the tidal cycle and one could call this a bottom pool.

2.      Line 77: "in over" → "over" (typo)

done

3.	Line 195-196: "were added on the bed", perhaps extend to "were added on the bed, i.e. global erosion limited conditions (f=1)." To me, this would clarify the definition od sediment transport capacity as being the maximum sediment transport that is possible under local hydrodynamic conditions.

This would be one definition. Another definition that one could use is how the total transport would change if a small uniform layer of sediment is added to the bottom pool. The latter is the definition we were thinking of, since this does not specify how much sediment there should be on the bed. Hence, this also works in the nonlinear Loire case.

4.	Line 262-263: "For alle model settings … refer to DgD24". I would give the authors the consideration to include the settings in an Appendix. After all, DGD24 has not yet been accepted yet…

Settings have been added to an Appendix. We expect that the paper of DdG24 will be accepted very soon.

5.	Line 400-402: The downward zero-crossing at 8 km is not well visible during neap tide.

We agree, but it is there

6.	Line 402-403: "This helps to explain the quite minor differences between spring and neap … noted earlier.". I think this should be minor *qualitative* differences as the magnitude of the concentration varies by a factor two.

done

7.	Line 403-405: To what extend can hindered settling also contribute to sediments being kept higher in the water column and thus contribute to the increased sediment capacity?

This is a significant contribution. At the maximum concentration around 40 g/l, hindered settling reduces the effective settling velocity to about 10% of the clear water value. Reduction of the eddy viscosity is of a similar order of magnitude. Hindered settling has been added to the text here.

8.	Line 418-419: "Erodibility is nevertheless … and strong stratification." To me, this may be seen as mimicking a situation of low erosion parameter M which indeed corresponds to a greater likelihood of erosion limitation (see Fig. 5).

If only focussed on the sediment capacity, changing M has a similar effect indeed. Please note though that the change in the tidal amplitude not only changes the sediment capacity but also the transport capacity. A change in M would not have the same effect.

9.	Line 420-421: "…, where the quasi-stationary case shows flushing of the bottom pool. In the dynamic case, the bottom pool remains present". I agree that the bottom pool is flushed in the quasi-stationary case. However, as it it written here I interpret the remark about the dynamic case as that the bottom pool is *always* present which is not correct. Indeed, f=1 occurs only after neap at the downstream located ETM. Please explain that there is a permanent mud pool or restate this remark.

Our intention was that 'bottom pool' indicates any clearly non-zero value of the erodibility. To avoid confusion we have reformulated to 'In the dynamic case, the erodibility remains significantly larger than zero.'

10.     Line 420-423: I agree that sediment is transported to the bottom pool at the entrance at neap. I also agree that at spring there is transport towards the ETMs at 40 km, but I don't think there is a bottom pool there (f<1 at spring, see previous remark). I think that 'trapping areas' is more appropriate here. Besides, it also seems that there is accumulation of sediment at the 40 km ETM after neap.

As above, 'bottom pool' does not just indicate f=1. Also for f<1 there is sediment on the bed during some part of the tide, so it is fair to speak of a bottom pool. Here, we believe there could be no confusion about f=1 or f<1 since f<1 clearly at all time around km 40.

11.     Line 423: "..., sediment is transported between the two bottom pools between spring and neap". First, I only think there is one bottom pool (see previous remarks above). Second, I found this not so clear from what the authors wrote. I think that sediment is being transported from the downstream to the upstream ETM at 0.5<t/T<neap (blue region) and visa versa for 0<t/T<0.35 (red region). Is this what the authors want to convey? I would think that this back and forth transport is necessarily a recirculation (both net transports being equal) since the authors consider a situation that is equilibrium on the neap-spring timescale. Could the authors comment on this?

We have added the timing 0.5<t/T<1 towards the lower pool and 0<t/T<0.5 towards the upper pool. This is indeed what we wanted to say. Any back-and-forth transport should indeed be a recirculation because of the dynamic equilibrium. Our point here is that there is such a recirculation; one could alternatively think that the same trapping location means that there is no transport of sediment at all, which is not the case here.